# SynEM, automated synapse detection for connectomics

Benedikt Staffler[1], Manuel Berning[1], Kevin M Boergens[1], Anjali Gour[1], Patrick van der Smagt[2†], Moritz Helmstaedter[1]*

[1]Department of Connectomics, Max Planck Institute for Brain Research, Frankfurt, Germany; [2]Biomimetic Robotics and Machine Learning, Munich, Germany

**Abstract** Nerve tissue contains a high density of chemical synapses, about 1 per $\mu m^3$ in the mammalian cerebral cortex. Thus, even for small blocks of nerve tissue, dense connectomic mapping requires the identification of millions to billions of synapses. While the focus of connectomic data analysis has been on neurite reconstruction, synapse detection becomes limiting when datasets grow in size and dense mapping is required. Here, we report SynEM, a method for automated detection of synapses from conventionally en-bloc stained 3D electron microscopy image stacks. The approach is based on a segmentation of the image data and focuses on classifying borders between neuronal processes as synaptic or non-synaptic. SynEM yields 97% precision and recall in binary cortical connectomes with no user interaction. It scales to large volumes of cortical neuropil, plausibly even whole-brain datasets. SynEM removes the burden of manual synapse annotation for large densely mapped connectomes.

*For correspondence: mh@brain. mpg.de

Present address: †Data Lab, VW Group, Munich, Germany

Competing interests: The authors declare that no competing interests exist.

## Introduction

The ambition to map neuronal circuits in their entirety has spurred substantial methodological developments in large-scale 3-dimensional microscopy (*Denk and Horstmann, 2004*; *Hayworth et al., 2006*; *Knott et al., 2008*; *Eberle et al., 2015*), making the acquisition of datasets as large as 1 cubic millimeter of brain tissue or even entire brains of small animals at least plausible (*Mikula et al., 2012*; *Mikula and Denk, 2015*). Data analysis, however, is still lagging far behind (*Helmstaedter, 2013*). One cubic millimeter of gray matter in the mouse cerebral cortex, spanning the entire depth of the gray matter and comprising several presumed cortical columns (*Figure 1a*), for example, contains at least 4 kilometers of axons, about 1 kilometer of dendritic shafts, about 1 billion spines (contributing an additional 2–3 kilometers of spine neck path length) and about 1 billion synapses (*Figure 1b*). Initially, neurite reconstruction was so slow, that synapse annotation comparably paled as a challenge (*Figure 1c*): when comparing the contouring of neurites (proceeding at 200–400 work hours per millimeter neurite path length) with synapse annotation by manually searching the volumetric data for synaptic junctions (*Figure 1d*, proceeding at about 0.1 hr per $\mu m^3$), synapse annotation consumed at least 20-fold less annotation time than neurite reconstruction (*Figure 1c*). An alternative strategy for manual synapse detection is to follow reconstructed axons (*Figure 1e*) and annotate sites of vesicle accumulation and postsynaptic partners. This axon-focused synapse annotation reduces synapse annotation time by about 8-fold for dense reconstructions (proceeding at about 1 min per potential contact indicated by a vesicle accumulation, which occurs every about 4–10 $\mu m$ along axons in mouse cortex).

With the development of substantially faster annotation strategies for neurite reconstruction, however, the relative contribution of synapse annotation time to the total reconstruction time has substantially changed. Skeleton reconstruction (*Helmstaedter et al., 2011*) together with automated volume segmentations (*Helmstaedter et al., 2013*; *Berning et al., 2015*), allow to proceed at about

**eLife digest** Each nerve cell in the brain of a mammal communicates with about 1,000 other nerve cells in a complex network. Nerve cells 'talk' to each other via structures called synapses that connect the nerve cells together. The number of synapses in the brain is enormous – for example, a human brain contains about one quadrillion synapses.

One technique that can be used to look at the synapses in the brain is called 3D electron microscopy. The huge number of synapses in an image makes it impractical for researchers to manually label them. However, current methods that use computers to automatically label synapses work most accurately only on images that are so detailed that they cover only very small volumes of the brain (much less than 1 cubic millimeter).

Staffler et al. have now developed a new method, called SynEM, that makes it possible for computers to do all the work of finding the synapses in larger volumes of the brain. Without any input from researchers, SynEM can correctly identify connections between nerve cells 97% of the time, which is far more successful than any other current computer-based approach. Importantly, SynEM also automatically indicates which nerve cells are connected by a given synapse, providing a map of "who talks to whom" across the brain.

Together with SynEM, methods that track the cable-like structures (called neurites) that nerve cells grow to find other nerve cells are already allowing us to map the communication networks in the brain. In the far future Staffler et al. hope that such mappings will become so routine that entire human brains could be studied, perhaps to investigate how diseases affect them.

7–10 hr per mm path length (mouse retina, *Helmstaedter et al., 2013*) or 4–7 hr per mm (mouse cortex, *Berning et al., 2015*), thus about 50-fold faster than manual contouring. Recent improvements in online data delivery and visualization (*Boergens et al., 2017*) further reduce this by about 5–10 fold. Thus, synapse detection has become a limiting step in dense large-scale connectomics. Importantly, any further improvements in neurite reconstruction efficiency would be bounded by the time it takes to annotate synapses. Therefore, automated synapse detection for large-scale 3D EM data is critical.

High-resolution EM micrographs are the gold standard for synapse detection (*Gray, 1959*; *Colonnier, 1968*). Images acquired at about 2–4 nm in-plane resolution have been used to confirm chemical synapses using the characteristic intense heavy metal staining at the postsynaptic membrane, thought to be caused by the accumulated postsynaptic proteins ('postsynaptic density', PSD), and an agglomeration of synaptic vesicles at the membrane of the presynaptic terminal. While synapses can be unequivocally identified in 2-dimensional images when cut perpendicularly to the synaptic cleft (*Figure 1f*), synapses at oblique orientations or with a synaptic cleft in-plane to the EM imaging are hard or impossible to identify. Therefore, the usage of 3D EM imaging with a high resolution of 4–8 nm also in the cutting dimension (FIB/SEM, *Knott et al., 2008*) is ideal for synapse detection. For such data, automated synapse detection is available and successful (*Kreshuk et al., 2011*; *Becker et al., 2012*, *2013*, *Supplementary file 1*). However, FIB-SEM currently does not scale to large volumes required for connectomics of the mammalian cerebral cortex. Serial Blockface EM (SBEM, *Denk and Horstmann, 2004*) scales to such mm$^3$ -sized volumes. However, SBEM provides a resolution just sufficient to follow all axons in dense neuropil and to identify synapses across multiple sequential images, independent of synapse orientation (*Figure 1g*, see also Synapse Gallery in *Supplementary file 4*; the resolution of SBEM is typically about 10 x 10 × 30 nm$^3$; *Figure 1g*). In this setting, synapse detection methods developed for high-in plane resolution data do not provide the accuracy required for fully automated synapse detection (see below).

Here we report SynEM, an automated synapse detection method based on an automated segmentation of large-scale 3D EM data (using SegEM, *Berning et al., 2015*; an earlier version of SynEM was deposited on biorxiv, *Staffler et al., 2017*). SynEM is aimed at providing fully automated connectomes from large-scale EM data in which manual annotation or proof reading of synapses is not feasible. SynEM achieves precision and recall for single-synapse detection of 88% and for binary

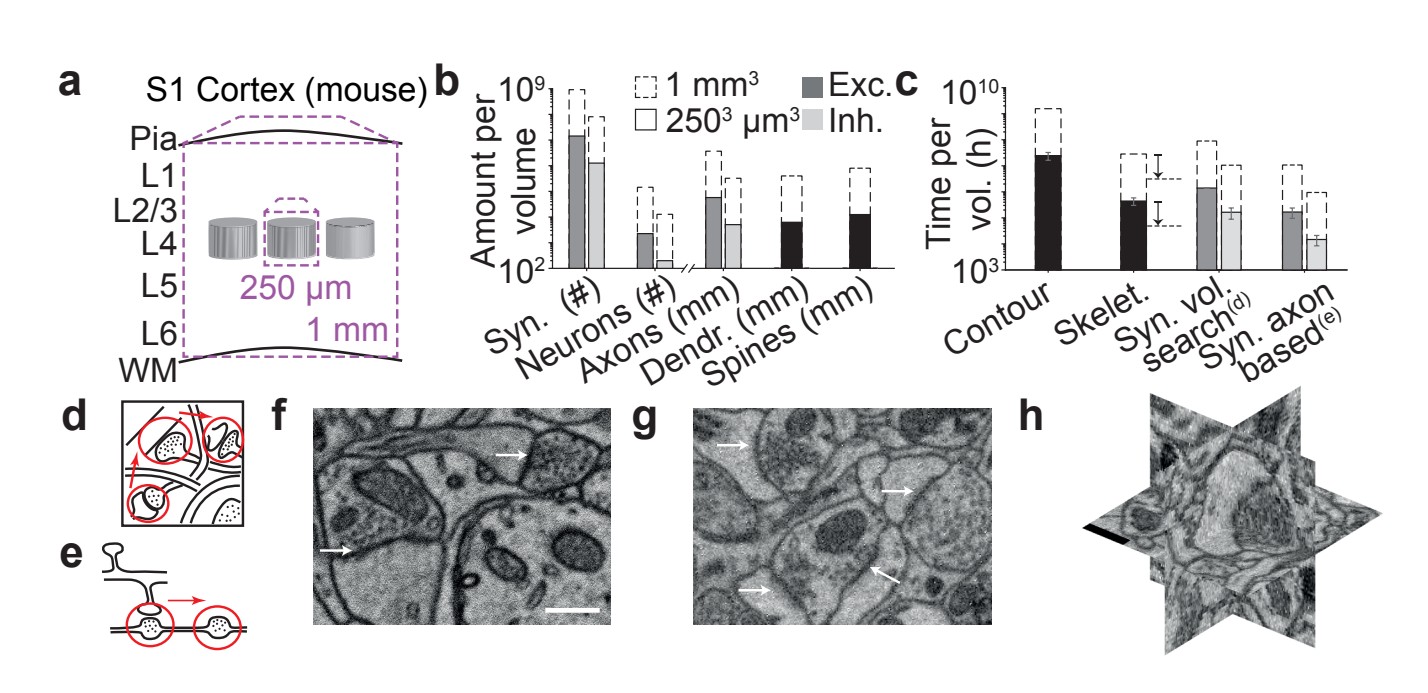

**Figure 1.** The challenge of synapse detection in connectomics. (**a**) Sketch of mouse primary somatosensory cortex (S1) with circuit modules ('barrels') in cortical layer 4 and minimum required dataset extent for a 'barrel' dataset (250 µm edge length) and a dataset extending over the whole cortical depth from pia to white matter (WM) (1 mm edge length). (**b**) Number of synapses and neurons, total axonal, dendritic and spine path length for the example datasets in (a) (*White and Peters, 1993*; *Braitenberg and Schüz, 1998*; *Merchán-Pérez et al., 2014*). (**c**) Reconstruction time estimates for neurites and synapses; For synapse search strategies see sketches in d,e. Dashed arrows: latest skeletonization tools (webKnossos, *Boergens et al., 2017*) allow for a further speed up of neurite skeletonization by about 5-to-10-fold, leaving synapse detection as the main annotation bottleneck. (**d**) Volume search for synapses by visually investigating 3d image stacks and keeping track of already inspected locations takes about 0.1 h/µm³. (**e**) Axon-based synapse detection by following axonal processes and detecting synapses at boutons consumes about 1 min per bouton. (**f**) Examples of synapses imaged at an in-plane voxel size of 6 nm and (**g**) 12 nm in conventionally en-bloc stained and fixated tissue (*Briggman et al., 2011*; *Hua et al., 2015*) imaged using SBEM (*Denk and Horstmann, 2004*). Arrows: synapse locations. Note that synapse detection in high-resolution data is much facilitated in the plane of imaging. Large-volume image acquisition is operated at lower resolution, requiring better synapse detection algorithms. (**h**) Synapse shown in 3D EM raw data, resliced in the 3 orthogonal planes. Scale bars in f and h, 500 nm. Scale bar in f applies to g.

The following source data is available for figure 1:

**Source data 1.** Source data for plots in panels 1b, 1c.

neuron-to-neuron connectomes of 97% without any human interaction, essentially removing the synapse annotation challenge for large-scale mammalian connectomes.

## Results

### Interface classification

We consider synapse detection as a classification of interfaces between neuronal processes as synaptic or non-synaptic (*Figure 2a*; see also *Mishchenko et al., 2010*, *Kreshuk et al., 2015*, *Huang et al., 2016*). This approach relies on a volume segmentation of the neuropil sufficient to provide locally continuous neurite pieces (such as provided by SegEM, *Berning et al., 2015*, for SBEM data of mammalian cortex), for which the contact interfaces can be evaluated.

The unique features of synapses are distributed asymmetrically around the synaptic interface: presynaptically, vesicle pools extend into the presynaptic terminal over at least 100–200 nm; postsynaptically, the PSD has a width of about 20–30 nm. To account for this surround information our classifier considers the subvolumes adjacent to the neurite interface explicitly and separately, unlike previous approaches (*Kreshuk et al., 2015*; *Huang et al., 2016*), up to distances of 40, 80, and 160

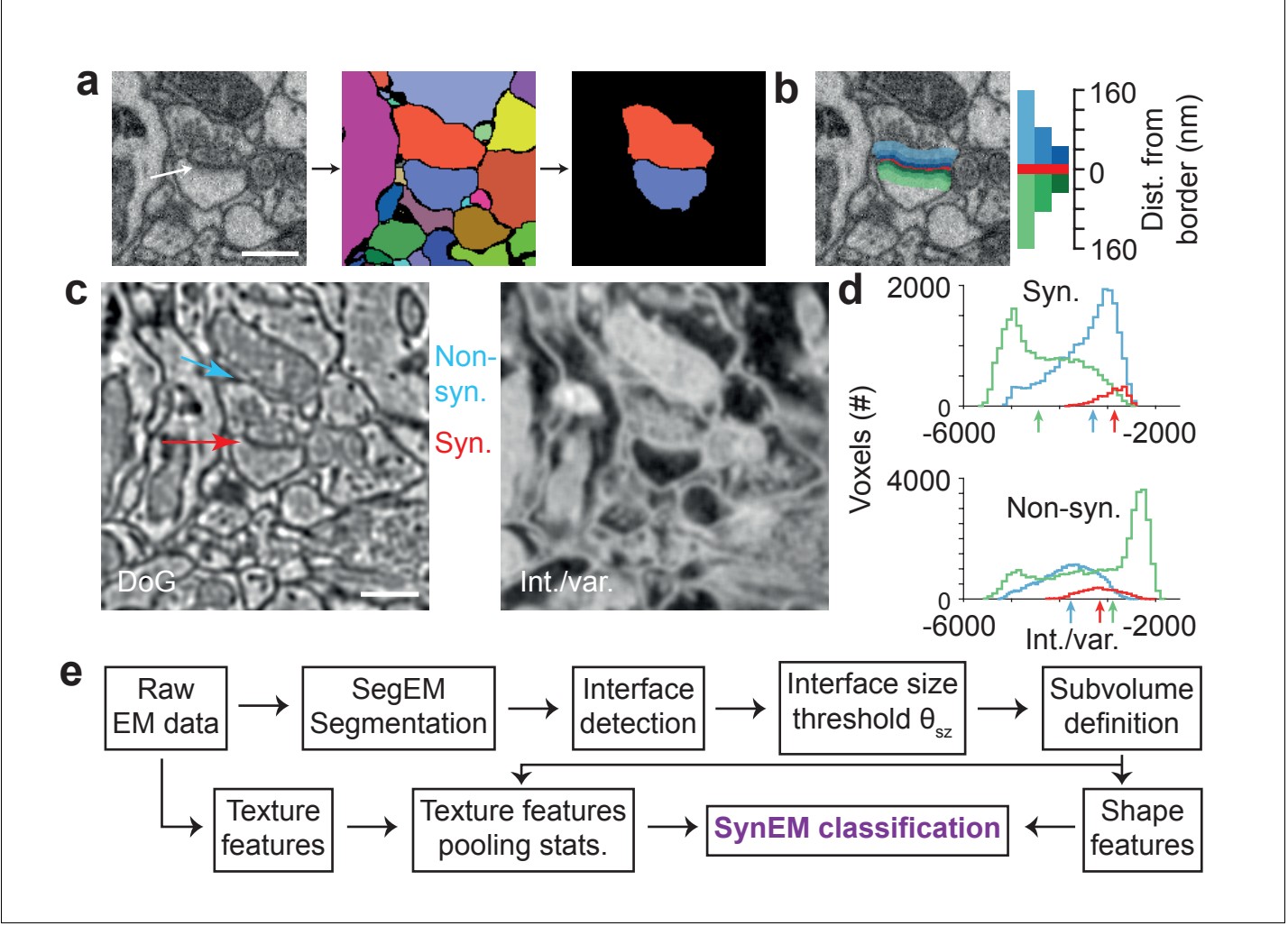

**Figure 2.** Synapse detection by classification of neurite interfaces. (**a**) Definition of interfaces used for synapse classification in SynEM. Raw EM data (left) is first volume segmented (using SegEM, *Berning et al., 2015*). Neighboring volume segments are identified (right). (**b**) Definition of perisynaptic subvolumes used for synapse classification in SynEM consisting of a border (red) and subvolumes adjacent to the neurite interface extending to distances of 40, 80 and 160 nm. (**c**) Example outputs of two texture filters: the difference of Gaussians (DoG) and the intensity/variance filter (int./var.). Note the clear signature of postsynaptic spine heads (right). (**d**) Distributions of int/var. texture filter output for image voxels at a synaptic (top) and non-synaptic interface (bottom). Medians over subvolumes are indicated (arrows, color scale as in b). (**e**) SynEM flow chart. Scale bars, 500 nm. Scale bar in a applies to a,b.

The following source data is available for figure 2:

**Source data 1.** Source data for plot in panel 2d.

nm from the interface, restricted to the two segments in question (*Figure 2b*; the interface itself was considered as an additional subvolume). We then compute a set of 11 texture features (*Table 1*, this includes the raw data as one feature), and derive 9 simple aggregate statistics over the texture features within the 7 subvolumes. In addition to previously used texture features (*Kreshuk et al., 2011*, *Table 1*), we use the local standard deviation, an intensity-variance filter and local entropy to account for the low-variance ('empty') postsynaptic spine volume and presynaptic vesicle clouds, respectively (see *Figure 2c* for filter output examples and *Figure 2d* for filter distributions at an example synaptic and non-synaptic interface). The 'sphere average' feature was intended to provide information about mitochondria, which often impose as false positive synaptic interfaces when adjacent to a plasma membrane. Furthermore, we employ 5 shape features calculated for the border subvolume and the

**Table 1.** Overview of the classifier features used in SynEM, and comparison with existing methods. 11 3-dimensional texture filters employed at various filter parameters given in units of standard deviation (s) of Gaussian filters (s was 12/11.24 voxels in x and y-dimension and 12/28 voxels in z-dimension, sizes of filters were set to σ/s*ceil(2*s)). When structuring elements were used, $1_{axbxc}$ refers to a matrix of size a x b x c filled with ones and r specifies the semi-principal axes of an ellipsoid of length (r, r, r/2) voxels in x, y and z-dimension. All texture features are pooled by 9 summary statistics (quantiles (0.25, 0.5, 0.75, 0, 1), mean, variance, skewness, kurtosis, respectively) over the 7 subvolumes around the neurite interface (see **Figure 2b**). Shape features were calculated for three of the subvolumes: border (Bo) and the 160 nm distant pre- and postsynaptic volumes (160). Init. Class: initial SynEM classifier (see **Figure 3d** for performance evaluation). N of instances: number of feature instances per subvolume (n = 7) and aggregate statistic (n = 9). *: Total number of employed features is 63 times reported instances for texture features. For shape features, the reported number is the total number of instances used, together yielding 3224 features total.

| Features | Kreshuk et al. (2011) | Becker et al. (2012) | Init. class. | SynEM | Parameters | N of instances* |
|---|---|---|---|---|---|---|
| **Texture:** | | | | | | |
| Raw data | | × | × | × | - | 1 |
| 3 EVs of Structure Tensor | × | × | × | × | $(\sigma_w, \sigma_d)$ = {(s,s), (s,2s), (2 s,s), (2 s,2s), (3 s,3s)} | 15 |
| 3 EVs of Hessian | × | × | × | × | σ = {s, 2 s, 3 s, 4 s} | 12 |
| Gaussian Smoothing | × | | × | × | σ = {s, 2 s, 3 s} | 3 |
| Difference of Gaussians | × | | | × | (σ,k) = {(s, 1.5), (s, 2), (2 s, 1.5), (2 s, 2), (3 s, 1.5)} | 5 |
| Laplacian of Gaussian | × | × | × | × | σ = {s, 2 s, 3 s, 4 s} | 4 |
| Gauss Gradient Magn. | × | × | × | × | σ = {s, 2 s, 3 s, 4 s, 5 s} | 5 |
| Local standard deviation | | | | × | U = $1_{5x5x5}$ | 1 |
| Int./var. | | | | × | U = {$1_{3x3x3}$, $1_{5x5x5}$} | 2 |
| Local entropy | | | | × | U = $1_{5x5x5}$ | 1 |
| Sphere average | | | | × | r = {3, 6} | 2 |
| **Shape:** | | | | | | |
| Number of voxels | | | × | × | Bo, 160 | 3 |
| Diameter (vx based) | | | | × | Bo | 1 |
| Lengths of principal axes | | | | × | Bo | 3 |
| Principal axis product | | | | × | 160 | 1 |
| Convex hull (vx based) | | | | × | Bo, 160 | 3 |

two subvolumes extending 160 nm into the pre- and postsynaptic processes, respectively. Together, the feature vector for classification had 3224 entries for each interface (**Table 1**).

## SynEM workflow and training data

We developed and tested SynEM on a dataset from layer 4 (L4) of mouse primary somatosensory cortex (S1) acquired using SBEM (dataset 2012-09-28_ex145_07x2, Boergens et al., unpublished; the dataset was also used in developing SegEM, **Berning et al., 2015**). The dataset had a size of 93 × 60 × 93 μm³ imaged at a voxel size of 11.24 × 11.24 × 28 nm³. The dataset was first volume segmented (SegEM, **Berning et al., 2015**, **Figure 2a**, see **Figure 2e** for a SynEM workflow diagram). Then, all interfaces between all pairs of volume segments were determined, and the respective subvolumes were defined. Next, the texture features were computed on the entire dataset and aggregated as described above. Finally, the shape features were computed. Then, the SynEM classifier was implemented to output a synapse score for each interface and each of the two possible pre-to-postsynaptic directions (**Figure 3a–c**). The SynEM score was then thresholded to obtain an automated classification of interfaces into synaptic / non-synaptic (θ in **Figure 3a**). Since the SynEM scores for the two possible synaptic directions at a given neurite-to-neurite interface were rather disjunct in the range of relevant thresholds, we used the larger of the two scores for classification

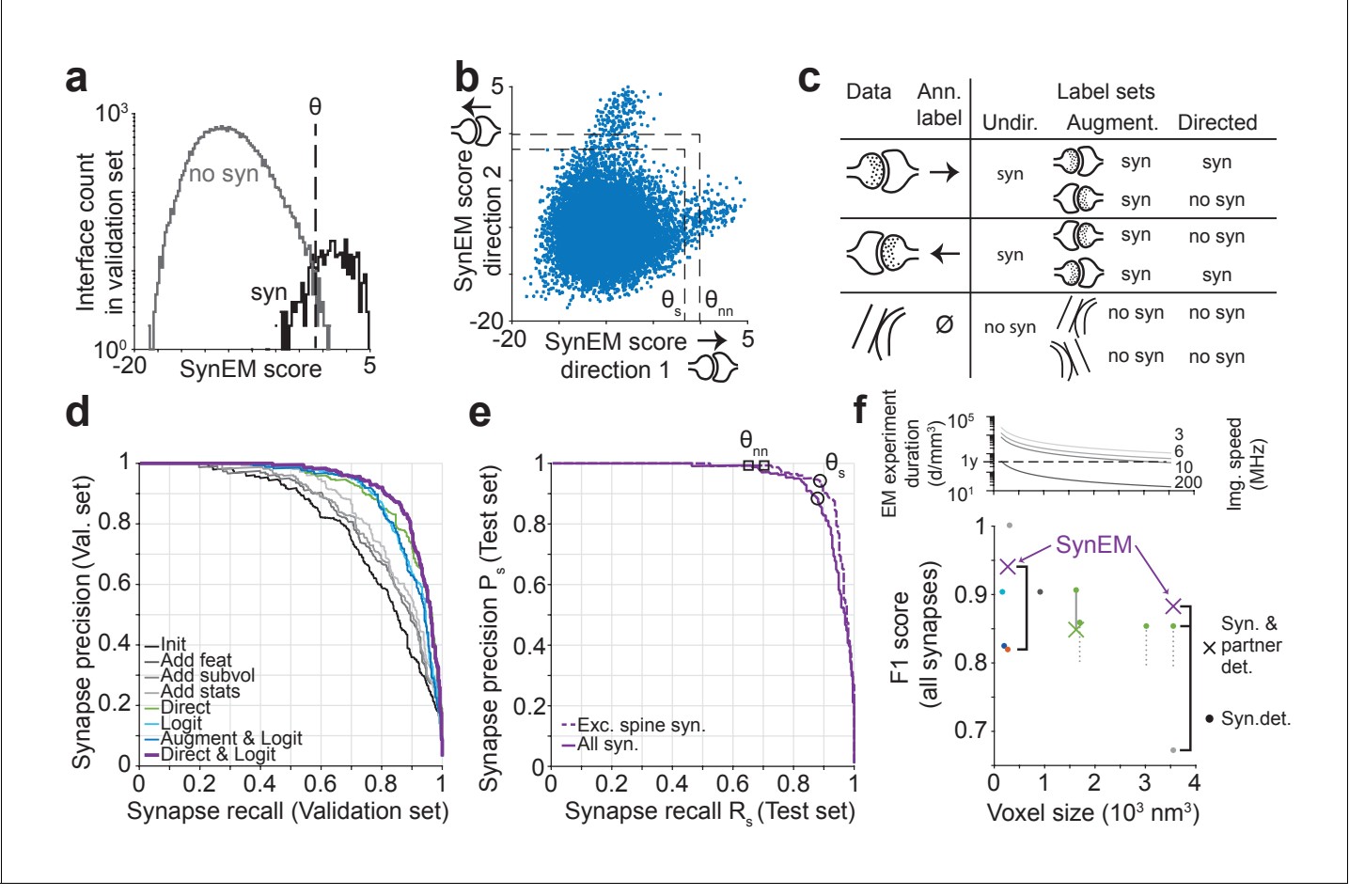

**Figure 3.** SynEM training and evaluation. (**a**) Histogram of SynEM scores calculated on the validation set. Fully automated synapse detection is obtained by thresholding the SynEM score at threshold θ. (**b**) SynEM scores for the two possible directions of interfaces. Note that SynEM scores are disjunct in a threshold regime used for best single synapse performance (θₛ) and best neuron-to-neuron recall and precision (θₙₙ), see *Figure 5*, indicating a clear bias towards one of the two possible synaptic directions. (**c**) Strategy for label generation. Based on annotator labels (Ann. Label), three types of label sets were generated: Initial label set ignored interface orientation (Undir.); Augmented label set included mirror-reflected interfaces (Augment.); Directed label set used augmented data but considered only one synaptic direction as synaptic (Directed, see also *Figure 3—figure supplement 1*). (**d**) Development of the SynEM classifier. Classification performance for different features, aggregation statistics, classifier parameters and label sets. Init: initial classifier used (see *Table 1*). The initial classifier was extended by using additional features (Add feat, see *Table 1*, first row), 40 and 80 nm subvolumes for feature aggregation (Add subvol, see *Figure 2b*) and aggregate statistics (Add stats, see *Table 1*). Direct: Classifier trained on directed label set (see *Figure 3c*). Logit: Classifier trained on full feature space using LogitBoost. Augment and Logit: Logit classifier trained on augmented label set (see *Figure 3c*). Direct and Logit: Logit classifier trained on directed label set (see *Figure 3c*). (**e**) Test set performance on 3D SBEM data of SynEM (purple) evaluated for spine and shaft synapses (all synapses, solid line) and for spine synapses (exc. synapses, dashed line), only. Threshold values for optimal single synapse detection performance (black circle) and an optimal connectome reconstruction performance (black square, see *Figure 5*). (see also *Figure 3—figure supplement 2*) (**f**) Relation between 3D EM imaging resolution, imaging speed and 3D EM experiment duration (top), exemplified for a dataset sized 1 mm³. Note that the feasibility of experiments strongly depends on the chosen voxel size. Bottom: published synapse detection performance (reported as F1 score) in dependence of the respective imaging resolution (see also *Supplementary file 1*). dark blue, *Mishchenko et al. (2010)*; cyan, *Kreshuk et al. (2011)*; light gray, *Becker et al. (2012)*; dark gray, *Kreshuk et al. (2014)*; red, *Roncal et al. (2015)*; green, *Dorkenwald et al. (2017)*; Black brackets indicate direct comparison of SynEM to top-performing methods: SynEM vs *Roncal et al. (2015)* on ATUM-SEM dataset (*Kasthuri et al., 2015*); SynEM vs *Dorkenwald et al. (2017)* and *Becker et al. (2012)* on our test set. See *Figure 3—figure supplement 3* for comparison of Precision-Recall curves. Note that SynEM outperforms the previously top-performing methods. Note also that most methods provide synapse detection, but require the detection of synaptic partners and synapse direction in a separate classification step. Gray solid line: drop of partner detection performance compared to synapse detection in *Dorkenwald et al. (2017)*; dashed gray lines, analogous possible range of performance drop as reported for bird dataset in *Dorkenwald et al. (2017)*. SynEM combines synapse detection and partner detection into one classification step.

The following source data and figure supplements are available for figure 3:

*Figure 3 continued on next page*

*Figure 3 continued*

**Source data 1.** Source data for plots in panels 3a, 3b, 3d, 3e, 3f.
**Figure supplement 1.** Graphical user interface (implemented in MATLAB) for efficient annotation of neurite interfaces as used for generating the training and validation labels.
**Figure supplement 2.** Distribution of training, validation and test data volumes within the dataset 2012-09-28_ex145_07x2.
**Figure supplement 3.** Synapse detection performance comparison of SynEM with SyConn (*Dorkenwald et al., 2017*) and (*Becker et al., 2012*) on the 3D SBEM SynEM test set (*Figure 3e*).
**Figure supplement 3—source data 1.**
**Figure supplement 4.** Synapse detection performance comparison of SynEM with VesicleCNN (*Roncal et al., 2015*) on a 3D EM dataset from mouse S1 cortex obtained using ATUM-SEM (*Kasthuri et al., 2015*).
**Figure supplement 4—source data 2.**

(*Figure 3b*; $\theta_s$ and $\theta_{nn}$ refer to the SynEM thresholds optimized for single synapse or neuron-to-neuron connectome reconstruction, respectively, see below).

We obtained labels for SynEM training and validation by presenting raw data volumes of $(1.6 \times 1.6 \times 0.7{-}1.7)$ µm$^3$ that surrounded the segment interfaces to trained student annotators (using a custom-made annotation interface in Matlab, *Figure 3—figure supplement 1*). The raw data were rotated such that the interface was most vertically oriented in the image plane presented to the annotators; the two interfacing neurite segments were colored transparently for identification (this could be switched off by the annotators when inspecting the synapse, see Materials and methods for details). Annotators were asked to categorize the presented interface as either non-synaptic, pre-to-postsynaptic, or post-to-presynaptic (*Figure 3c*, *Figure 3—figure supplement 1*). The synaptic labels were then verified by an expert neuroscientist. A total of 75,383 interfaces (1858 synaptic, 73,525 non-synaptic) were annotated in image volumes drawn from 40 locations within the entire EM dataset (*Figure 3—figure supplement 2*). About 80% of the labels (1467 synaptic, 61,619 non-synaptic) were used for training, the remaining were used for validation.

Initially, we interpreted the annotator's labels in an undirected fashion: irrespective of synapse direction, the label was interpreted as synaptic (and non-synaptic otherwise, *Figure 3c*, 'Undir.'). We then augmented the training data by including mirror-reflected copies of the originally presented synapses, maintaining the labels as synaptic (irrespective of synapse direction) and non-synaptic (*Figure 3c*, 'Augmented'). Finally, we changed the labels of the augmented training data to reflect the direction of synaptic contact: only synapses in one direction were labeled as synaptic, and non-synaptic in the inverse direction (*Figure 3c*, 'Directed').

## SynEM evaluation

*Figure 3d* shows the effect of the choice of features, aggregate statistics, classifier parameters and label types on SynEM precision and recall. Our initial classifier used the texture features from *Kreshuk et al. (2011)* with minor modifications and in addition the number of voxels of the interface and the two interfacing neurite segmentation objects (restricted to 160 nm distance from the interface) as a first shape feature (*Table 1*). This classifier provided only about 70% precision and recall (*Figure 3d*). We then extended the feature space by adding more texture features capturing local image statistics (*Table 1*) and shape features. In particular, we added filters capturing local image variance in an attempt to represent the 'empty' appearance of postsynaptic spines, and the presynaptic vesicle clouds imposing as high-frequency high-variance features in the EM images. Also, we added more subvolumes over which features were aggregated (see *Figure 2b*), increasing the dimension of the feature space from 603 to 3224. Together with additional aggregate statistics, the classifier reached about 75% precision and recall. A substantial improvement was obtained by switching from an ensemble of decision-stumps (one-level decision tree) trained by AdaBoostM1

(*Freund and Schapire, 1997*) as classifier to decision stumps trained by LogitBoost (*Friedman et al., 2000*). In addition, the directed label set proved to be superior. Together, these improvements yielded a precision and recall of 87% and 86% on the validation set (*Figure 3d*).

We then evaluated the best classifier from the validation set (*Figure 3d*, 'Direct and Logit') on a separate test set. This test set was a dense volume annotation of all synapses in a randomly positioned region containing dense neuropil of size $5.8 \times 5.8 \times 7.2\ \mu m^3$ from the L4 mouse cortex dataset. All synapses were identified by two experts, which included the reconstruction of all local axons, and validated once more by another expert on a subset of synapses. In total, the test set contained 235 synapses and 20319 non-synaptic interfaces. SynEM automatically classified these at 88% precision and recall (*Figure 3e*, F1 score of 0.883). Since the majority of synapses in the cortex are made onto spines we also evaluated SynEM on all spine synapses in the test set (n = 204 of 235 synapses, 87%, *Figure 3e*). On these, SynEM performed even better, yielding 94% precision and 89% recall. (*Figure 3e*, F1 score of 0.914).

## Comparison to previous methods

We next compared SynEM to previously published synapse detection methods (*Figure 3f*, *Mishchenko et al., 2010*; *Kreshuk et al., 2011*, *2014*; *Becker et al., 2012*; *Roncal et al., 2015*; *Dorkenwald et al., 2017*). Other published methods were either already shown to be inferior to one of these approaches (*Perez et al., 2014*; *Márquez Neila et al., 2016*) or developed for specific subtypes of synapses, only (*Jagadeesh et al., 2014*; *Plaza et al., 2014*; *Huang et al., 2016*); these were

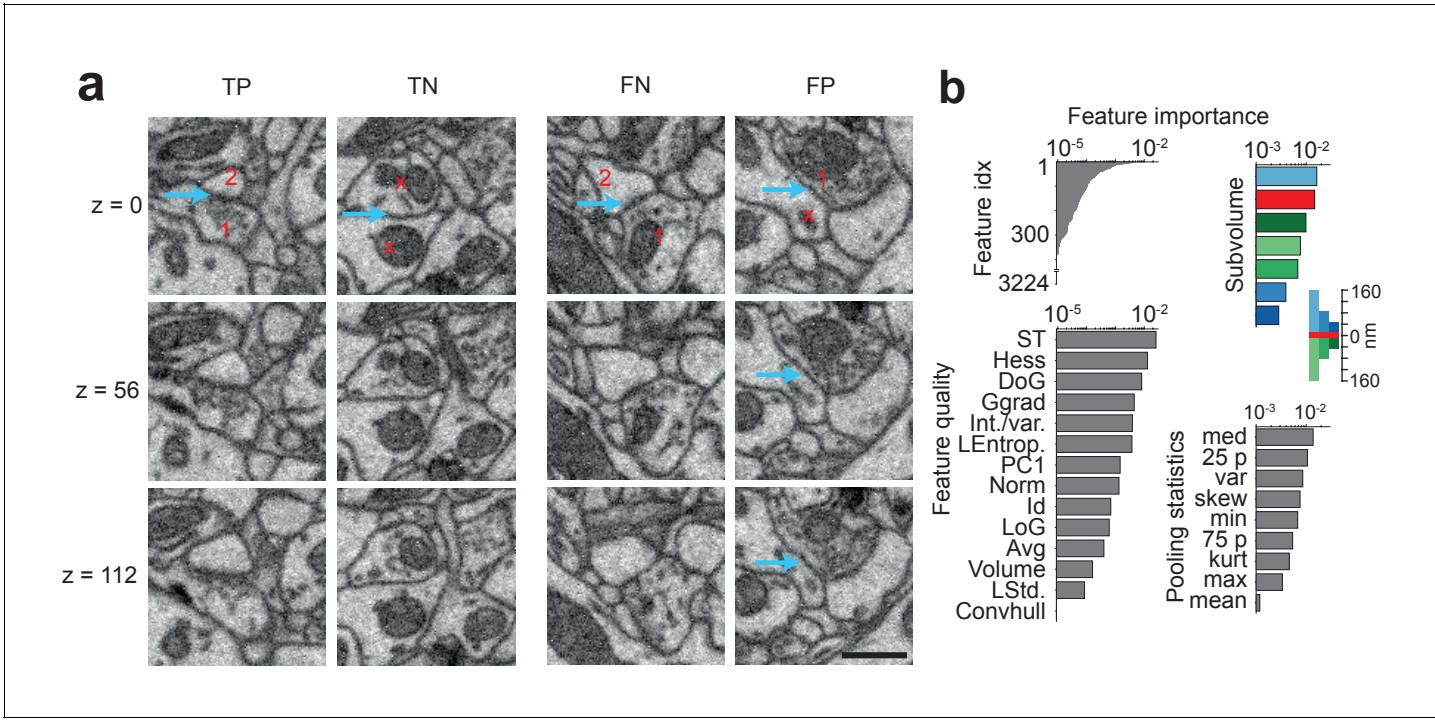

**Figure 4.** SynEM classification and feature importance. (**a**) SynEM classification examples at $\theta_s$ (circle in *Figure 3e*). True positive (TP), true negative (TN), false negative (FN) and false positive (FP) interface classifications (blue arrow, classified interface) shown as 3 image planes spaced by 56 nm (i.e. every second SBEM data slice, top to bottom). Note that synapse detection in 3D SBEM data requires inspection of typically 10–20 consecutive image slices (see Synapse Gallery in *Supplementary file 4* for examples). 1: presynaptic; 2: postsynaptic; x: non-synaptic. Note for the FP example that the axonal bouton (1) innervates a neighboring spine head, but the interface to the neurite under classification (x) is non-synaptic (blue arrow). (**b**) Ranked classification importance of SynEM features. All features (top left), relevance of feature quality (bottom left), subvolumes (top right) and pooling statistics (bottom right). Note that only 378 features contribute to classification. See *Table 2* for the 10 feature instances of highest importance, *Table 1* for feature name abbreviations, and text for details. Scale bars, 500 nm.

The following source data is available for figure 4:

**Source data 1.** Source data for plot in panel 4b.

therefore not included in the comparison. SynEM outperforms the state-of-the-art methods when applied to our SBEM data acquired at 3537 nm³ voxel size (*Figure 3f*, *Figure 3—figure supplement 3*). In addition, we applied SynEM to a published 3D EM dataset acquired at more than 10-fold smaller voxel size (3 × 3 × 30 = 270 nm³) using automated tape-collecting ultramicrotome-SEM imaging (ATUM, *Kasthuri et al., 2015*). SynEM also outperforms the method developed for this data (VesicleCNN, *Roncal et al., 2015*; *Figure 3f* and *Figure 3—figure supplement 4*), indicating that SynEM is applicable to EM data of various modalities and resolution.

It should furthermore be noted that for connectomics, in addition to the detection of the location of a synapse, the two neuronal partners that form the synapse and the direction of the synapse have to be determined. The performance of the published methods as reported in *Figure 3f* only include the synapse detection step. Interestingly, the recently published method (*Dorkenwald et al., 2017*) reported that the additional detection of the synaptic partners yielded a drop of performance of 2% precision and 9% recall (F1 score decreased by about 5% from 0.906 to 0.849) compared to synapse detection alone (*Figure 3f*, see *Dorkenwald et al., 2017*). This indicates that the actual performance of this method on our data would be lower when including partner detection. SynEM, because of the explicit classification of directed neurite interfaces, in contrast, explicitly provides synapse detection, partner detection and synapse directionality in one classification step.

## Remaining SynEM errors, feature importance, and computational feasibility

*Figure 4a* shows examples of correct and incorrect SynEM classification results (evaluated at $\theta_s$). Typical sources of errors are vesicle clouds close to membranes that target nearby neurites (*Figure 4a*, FP), Mitochondria in the pre- and/or postsynaptic process, very small vesicle clouds and/or small PSDs (*Figure 4a*, FN), and remaining SegEM segmentation errors. To estimate the effect of segmentation errors on SynEM performance, we investigated all false positive and false negative detections in the test set and checked for the local volume segmentation quality. We found that, in fact, 26 of the 28 FNs and 22 of the 27 FPs were at locations with a SegEM error in proximity. Correcting these errors also corrected the SynEM errors in 22 of 48 (46%) of the cases. This indicates that further improvement of volume segmentation can yield an even further reduction of the remaining errors in SynEM-based automated synapse detection.

**Table 2.** SynEM features ranked by ensemble predictor importance. See *Figure 4b* and Materials and methods for details. Note that two of the newly introduced features and one of the shape features had high classification relevance (Local entropy, Int./var., Principal axes length; cf. *Table 1*).

| Rank | Feature | Parameters | Subvolume | Aggregate statistic |
|---|---|---|---|---|
| 1 | EVs of Struct. Tensor (largest) | $\sigma_w = 2s$, $\sigma_D = s$ | 160 nm, S1 | Median |
| 2 | EVs of Struct. Tensor (smallest) | $\sigma_w = 2s$, $\sigma_D = s$ | 160 nm, S1 | Median |
| 3 | Local entropy | $U = 1_{5 \times 5 \times 5}$ | 160 nm, S2 | Variance |
| 4 | Difference of Gaussians | $\sigma = 3 s$, $k = 1.5$ | Border | $25^{th}$ perc |
| 5 | Difference of Gaussians | $\sigma = 2 s$, $k = 1.5$ | Border | Median |
| 6 | EVs of Struct. Tensor (middle) | $\sigma_w = 2s$, $\sigma_D = s$ | 40 nm, S2 | Min |
| 7 | Int./var. | $U = 1_{3 \times 3 \times 3}$ | Border | $75^{th}$ perc |
| 8 | EVs of Struct. Tensor (largest) | $\sigma_w = 2s$, $\sigma_D = s$ | 80 nm, S1 | $25^{th}$ perc |
| 9 | Gauss gradient magnitude | $\sigma = s$ | 40 nm, S2 | $25^{th}$ perc |
| 10 | Principal axes length (2nd) | - | Border | - |

**Table 3.** SynEM score thresholds and associated precision and recall. SynEM score thresholds θ chosen for optimized single synapse detection ($\theta_s$) and optimized neuron-to-neuron connection detection ($\theta_{nn}$) with respective single synapse precision ($P_s$) and recall ($R_s$) and estimated neuron-to-neuron precision and recall rates ($P_{nn}$, $R_{nn}$, respectively) for connectome binarization thresholds of $\gamma_{nn} = 1$ and $\gamma_{nn} = 2$ (see **Figure 5**).

| Threshold score | Single synapse $P_s/R_s$ | Neuron-to-neuron $P_{nn}/R_{nn}$ | |
|---|---|---|---|
| | | $\gamma_{nn} = 1$ | $\gamma_{nn} = 2$ |
| $\theta_s = -1.67$ (exc) | 88.5%/88.1% | 72.5%/99.7% | 98.1%/95.6% |
| $\theta_{nn} = -0.08$ (exc) | 99.4%/65.1% | 98.5%/97.1% | 100%/83.4% |
| $\theta_s = -2.06$ (inh) | 82.1%/74.9% | 77.1%/100% | 92.7%/99.5% |
| $\theta_{nn} = -1.58$ (inh) | 88.6%/67.8% | 84.7%/99.9% | 97.3%/98.5% |

We then asked which of the SynEM features had highest classification power, and whether the newly introduced texture and shape features contributed to classification. Boosted decision-stump classifiers allow the ranking of features according to their classification importance (**Figure 4b**). 378 out of 3224 features contributed to classification (leaving out the remaining features did not reduce accuracy). The 10 features with highest discriminative power (**Table 2**) in fact contained two of the added texture filters (int-var and local entropy) and a shape feature. The three most distinctive subvolumes (**Figure 4b**) were the large presynaptic subvolume, the border and the small postsynaptic subvolume. This suggests that the asymmetry in pre- vs. postsynaptic aggregation volumes in fact contributed to classification performance, with a focus on the presynaptic vesicle cloud and the postsynaptic density.

Finally, SynEM is sufficiently computationally efficient to be applied to large connectomics datasets. The total runtime on the 384592 $\mu m^3$ dataset was 2.6 hr on a mid-size computational cluster (480 CPU cores, 16 GB RAM per core). This would imply a runtime of 279.9 days for a large 1 $mm^3$ dataset, which is comparable to the time required for current segmentation methods, but much faster than the currently required human annotation time ($10^5$ to $10^6$ hr, **Figure 1c**). Note that SynEM was not yet optimized for computational speed (plain matlab code, see git repository posted at https://gitlab.mpcdf.mpg.de/connectomics/SynEM).

## SynEM for connectomes

We so far evaluated SynEM on the basis of the detection performance of single synaptic interfaces. Since we are interested in measuring the connectivity matrices of large-scale mammalian cortical circuits (connectomes) we obtained a statistical estimate of connectome error rates based on synapse detection error rates. We assume that the goal is a binary connectome containing the information whether pairs of neurons are connected or not. Automated synapse detection provides us with weighted connectomes reporting the number of synapses between neurons, from which we can obtain binary connectomes by considering all neuron pairs with at least $\gamma_{nn}$ synapses as connected (**Figure 5a**). Synaptic connections between neurons in the mammalian cerebral cortex have been found to be established via multiple synapses per neuron pair (**Figure 5b**, **Feldmeyer et al., 1999**, **2002**, **2006**; **Frick et al., 2008**; **Markram et al., 1997**, range 1–8 synapses per connection, mean 4.3 ± 1.4 for excitatory connections, **Supplementary file 2**). The effect of synapse recall $R_s$ on recall of neuron-to-neuron connectivity $R_{nn}$ can be estimated (**Figure 5c**) for each threshold $\gamma_{nn}$ given the distribution of the number of synapses per connected neuron pair $n_{syn}$. For connectomes in which neuron pairs with at least one detected synapse are considered as connected ($\gamma_{nn} = 1$), a neuron-to-neuron connectivity recall $R_{nn}$ of 97% can be achieved with a synapse detection recall $R_s$ of 65.1% (**Figure 5c**, black arrow) if synapse detection is independent between multiple synapses of the same neuron pair. SynEM achieves 99.4% synapse detection precision $P_s$ at this recall (**Figure 3e**).

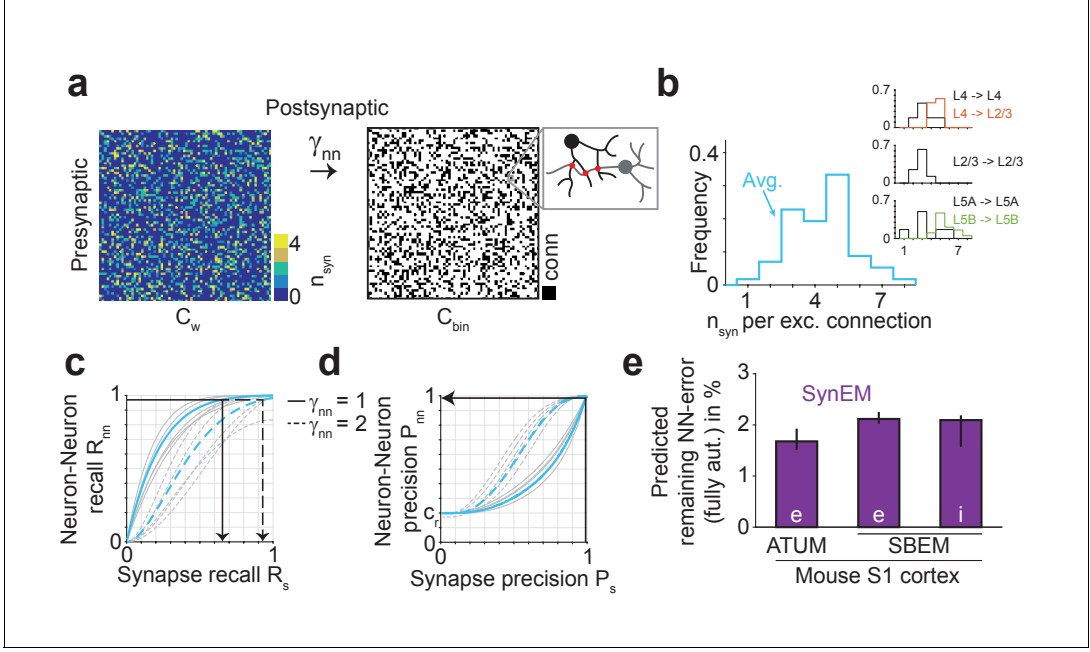

**Figure 5.** Effect of SynEM classification performance on error rates in automatically mapped binary connectomes. (**a**) Sketch of a weighted connectome (left) reporting the number of synapses per neuron-to-neuron connection, transformed into a binary connectome (middle) by considering neuron pairs with at least $\gamma_{nn}$ synapses as connected. (**b**) Distribution of reported synapse number for connected excitatory neuron pairs obtained from paired recordings in rodent cerebral cortex (*Feldmeyer et al., 1999*, *2002*, *2006*; *Frick et al., 2008*; *Markram et al., 1997*). Average distribution (cyan) is used for the precision estimates in the following (see *Supplementary file 2*). (**c**) Relationship between SynEM recall for single interfaces (synapses) $R_s$ and the ensuing neuron-to-neuron connectome recall $R_{nn}$ (recall in $C_{bin}$, a) for each of the excitatory cortico-cortical connections (summarized in b) and for connectome binarization thresholds of $\gamma_{nn} = 1$ and $\gamma_{nn} = 2$ (full and dashed, respectively). (**d**) Relationship between SynEM precision for single interfaces (synapses) $P_s$ and the ensuing neuron-to-neuron connectome precision $P_{nn}$. Colors as in c. (for inhibitory synapses see also *Figure 5—figure supplement 1*) (**e**) Predicted remaining error in the binary connectome (reported as 1-F1 score for neuron-to-neuron connections) for fully automated synapse classification using SynEM on 3D EM data from mouse cortex using two different imaging modalities: ATUM-SEM (left, *Kasthuri et al., 2015*) and our data using SBEM (right). e,i: excitatory or inhibitory connectivity model (see b and Materials and methods) shown for $c_{re} = 20\%$ and $c_{ri} = 60\%$. Black lines indicate range for varying assumptions of pairwise connectivity rate $c_{re} = (5\%, 10\%, 30\%)$ (excitatory) and $c_{ri} = (20\%, 40\%, 80\%)$ (inhibitory). Note that SynEM yields a remaining error of close to or less than 2%, well below expected biological wiring noise, allowing for fully automated synapse detection in large-scale binary connectomes. See Suppl. *Figure 5—figure supplement 2* for comparison to previous synapse detection methods.

The following source data and figure supplements are available for figure 5:

**Source data 1.** Source data for plots in panels 5b, 5c, 5d, 5e.

**Figure supplement 1.** Performance of SynEM on a test set containing all interfaces between 3 inhibitory axons and all touching neurites (total of 9430 interfaces, 171 synapses).

**Figure supplement 1—source data 1.**

**Figure supplement 2.** Effect of synapse detection errors on predicted connectome error rates for competing methods.

**Figure supplement 2—source data 2.**

The resulting precision of neuron-to-neuron connectivity $P_{nn}$ then follows from the total number of synapses in the connectome $N_{syn} = N^2 \times c_r \times \langle n_{syn} \rangle$, with $c_r$ the pairwise connectivity rate, about 20% for local excitatory connections in cortex (*Feldmeyer et al., 1999*), $\langle n_{syn} \rangle$ the mean number of synapses per connection (4.3 ± 1.4, *Figure 5b*), and $N^2$ the size of the connectome. A fraction $R_s$ of these synapses is detected (true positive detections, TPs). The number of false positive (FP) synapse detections was deduced from TP and the synapse precision $P_s$ as $FP = TP \times (1-P_s)/P_s$, yielding $R_s \times N_{syn} \times (1-P_s)/P_s$ false positive synapse detections. These we assumed to be distributed randomly

on the connectome and estimated how often at least $\gamma_{nn}$ synapses fell into a previously empty connectome entry. These we considered as false positive connectome entries, whose rate yields the binary connectome precision $P_{nn}$ (see Materials and methods for details of the calculation). At $R_{nn}$ of 97.1%, SynEM yields a neuron-to-neuron connection precision $P_{nn}$ of 98.5% (*Figure 5d*, black arrow, *Figure 5e*; note that this result is stable against varying underlying connectivity rates $c_{re}$ = 5%..30%, see indicated ranges in *Figure 5e*).

For the treatment of inhibitory connections, we followed the notion that synapse detection performance could be optimized by restricting classifications to interfaces established by inhibitory axons (as we had analogously seen for restricting analysis to spine synapses above, *Figure 3e*). For this, we evaluated SynEM on a test set of inhibitory axons for which we classified all neurite contacts of these axons (171 synapses, 9430 interfaces). While the precision and recall for single inhibitory synapses is lower than for excitatory ones (75% recall, 82% precision, *Figure 5—figure supplement 1*, SynEM$^{(i)}_s$), the higher number of synapses per connected cell pair ($n^{(i)}_{syn}$ is on average about 6, *Supplementary file 3*, *Gupta et al. (2000)*; *Markram et al. (2004)*; *Koelbl et al. (2015)*; *Hoffmann et al. (2015)*) still yields

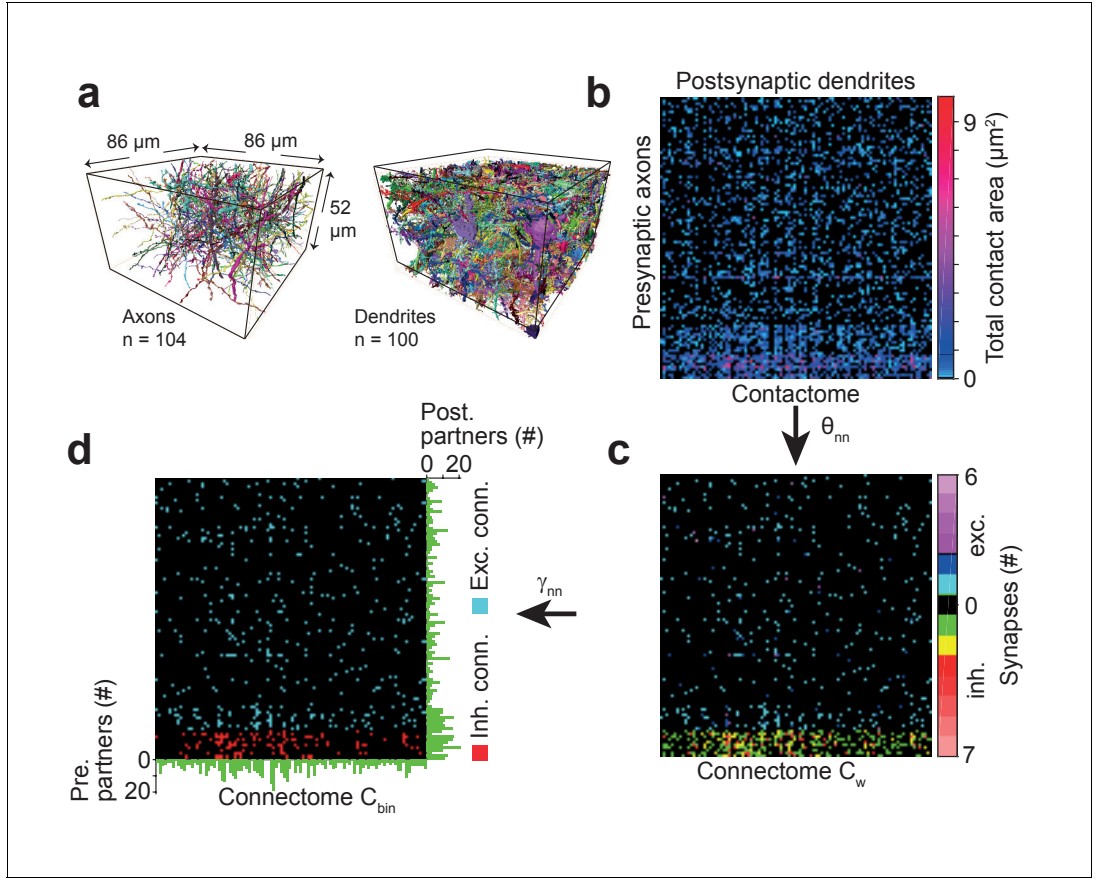

**Figure 6.** Example sparse local cortical connectome obtained using SynEM. (**a**) 104 axonal (94 excitatory, 10 inhibitory) and 100 dendritic processes within a volume sized 86 × 52 × 86 µm³ from layer 4 of mouse cortex skeletonized using webKnossos (*Boergens et al., 2017*), volume segmented using SegEM (*Berning et al., 2015*). (**b**) Contactome reporting total contact area between pre- and postsynaptic processes. (**c**) Weighted connectome obtained at the SynEM threshold $\theta_{nn}$ optimized for the respective presynaptic type (excitatory, inhibitory) (see *Figure 3e*, black square, *Table 3*). (see also *Figure 6—figure supplement 1*) (**d**) Binary connectome obtained from the weighted connectome by thresholding at $\gamma_{nn}$ = 1 for excitatory connections and $\gamma_{nn}$ = 2 for inhibitory connections. The resulting predicted neuron-to-neuron recall and precision were 98%, 98% for excitatory and 98%, 97% for inhibitory connections, respectively (see *Figure 5e*). Green: number of pre- (right) and postsynaptic (bottom) partners for each neurite.

The following source data and figure supplement are available for figure 6:

**Source data 1.** Source data for plots in panels 6b, 6c, 6d.

**Figure supplement 1.** Procedure for obtaining synapse counts in the local connectome (*Figure 6*).

substantial neuron-to-neuron precision and recall also for inhibitory connectomes (98% recall, 97% precision, *Figure 5e*, *Figure 5—figure supplement 1*, SynEM$^{(i)}_{nn}$; this result is stable against varying underlying inhibitory connectivity rates $c_{ri}$ = 20%..80%, see ranges indicated in *Figure 5e*). Error rates of less than 3% for missed connections and for wrongly detected connections are well below the noise of synaptic connectivity so far found in real biological circuits (e.g., *Helmstaedter et al., 2013*; *Bartol et al., 2015*), and thus likely sufficient for a large range of studies involving the mapping of cortical connectomes.

In summary, SynEM provides fully automated detection of synapses, their synaptic partner neurites and synapse direction for binary mammalian connectomes up to 97% precision and recall, a range which was previously prohibitively expensive to attain in large-scale volumes by existing methods (*Figure 5e*, *Figure 5—figure supplement 2*).

## Local cortical connectome

We applied SynEM to a sparse local cortical connectome between 104 axons and 100 postsynaptic processes in the dataset from L4 of mouse cortex (*Figure 6a*, neurites were reconstructed using webKnossos (*Boergens et al., 2017*) and SegEM as previously reported (*Berning et al., 2015*)). We first detected all contacts and calculated the total contact area between each pair of pre- and postsynaptic processes ('contactome', *Figure 6b*). We then classified all contacts using SynEM (at the classification threshold $\theta_{nn}$ (*Table 3*) yielding 98.5% precision and 97.1% recall for excitatory neuron-to-neuron connections and 97.3% precision and 98.5% recall for inhibitory neuron-to-neuron connections) to obtain the weighted connectome $C_w$ (*Figure 6c*). The detected synapses were clustered when they were closer than 1500 nm for a given neurite pair. This allowed us to concatenate large

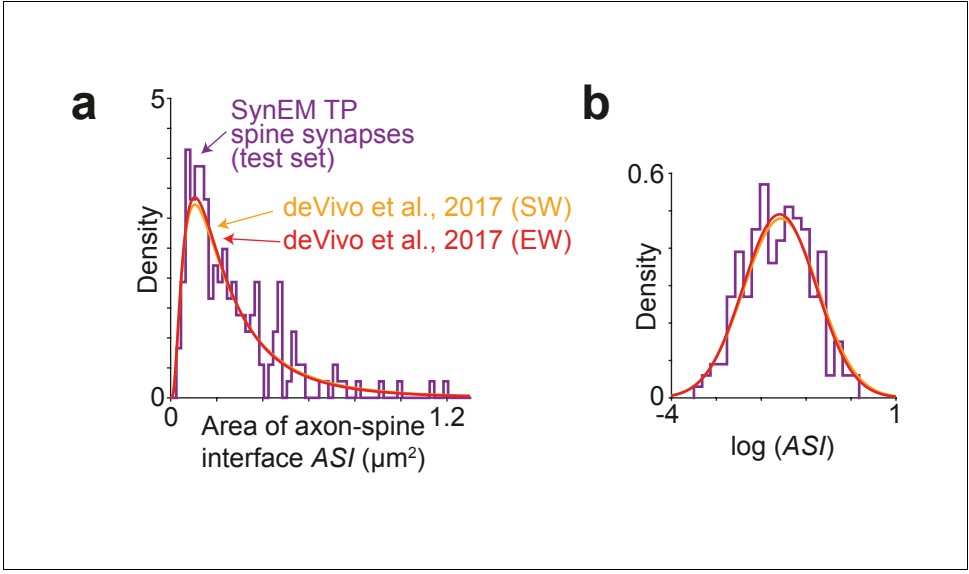

**Figure 7.** Comparison of synapse size in SBEM data. (**a**) Distribution of axon-spine interface area ASI for the SynEM-detected synapses onto spines in the test set from mouse S1 cortex imaged at 11.24 × 11.24 × 28 nm³ voxel size (see *Figure 3e*), purple; and distributions from *de Vivo et al. (2017)* in S1 cortex from mice under two wakefulness conditions (SW: spontaneous wake, EW: enforced wake), imaged at higher resolution of 5.9 nm (xy plane) with a section thickness of 54.7 ± 4.8 nm (SW), 51.4 ± 10.3 nm (EW) (*de Vivo et al., 2017*). (**b**) Same distributions as in (a) shown on natural logarithmic scale (log ASI SynEM −1.60 ± 0.74, n = 181; log ASI SW −1.56 ± 0.83, n = 839; log ASI EW −1.59 ± 0.81, n = 836; mean ± s.d.). Note that the distributions are indistinguishable (p=0.52 (SynEM vs. SW), p=0.83 (SynEM vs. EW), two-sample two-tailed t-test), indicating that the size distribution of synapses detected in our lower-resolution data is representative, and that SynEM does not have a substantial detection bias towards larger synapses.

The following source data is available for figure 7:

**Source data 1.** Source data for plots in panels 7a, 7b.

synapses with multiple active zones or multiple contributing SegEM segments into one (*Figure 6—figure supplement 1*). To obtain the binary connectome we thresholded the weighted connectome at $\gamma_{nn}$ = 1 for excitatory and at $\gamma_{nn}$ = 2 for inhibitory neuron-to-neuron connections (*Figure 6d*). The resulting connectome contained 880 synapses distributed over 536 connections.

## Frequency and size of automatically detected synapses

Finally, to check whether SynEM-detected synapses matched previous reports on synapse frequency and size, we applied SynEM to half of the entire cortex dataset used for this study (i.e. a volume of 192296 $\mu m^3$). SynEM detected 195644 synapses, i.e. a synapse density of 1.02 synapses per $\mu m^3$, consistent with previous reports (*Merchán-Pérez et al., 2014*).

We then measured the size of the axon-spine interface of SynEM detected synapses in the test set (*Figure 7a,b*). We find an average axon-spine interface size of 0.263 ± 0.206 $\mu m^2$ (mean ± s.d.; range 0.033–1.189 $\mu m^2$; n = 181), consistent with previous reports (*de Vivo et al., 2017*: (SW) 0.297 ± 0.297 $\mu m^2$ (p=0.518, two-sample two-tailed t-test on the natural logarithm of the axon-spine interface size), (EW) 0.284 ± 0.275 $\mu m^2$ (p=0.826, two-sample two-tailed t-test on the natural logarithm of the axon-spine interface size). This indicates that, first, synapse detection in our lower-resolution SBEM data (in-plane image resolution about 11 nm, section thickness about 26–30 nm) yields similar synapse size distributions as in the higher-resolution data in *de Vivo et al. (2017)* (in-plane image resolution 5.9 nm; section thickness about 50 nm) and secondly, that SynEM-based synapse detection has no obvious bias towards larger synapses.

## Discussion

We report SynEM, a toolset for automated synapse detection in EM-based connectomics. The particular achievement is that the synapse detection for densely mapped connectomes from the mammalian cerebral cortex is fully automated yielding below 3% residual error in the binary connectome. Importantly, SynEM directly provides the location and size of synapses, the involved neurites and the synapse direction without human interaction. With this, synapse detection is removed as a bottleneck in large-scale mammalian connectomics.

Evidently, synapse detection is facilitated in high-resolution EM data, and becomes most feasible in FIB-SEM data at a resolution of about 4–8 nm isotropic (*Kreshuk et al., 2011*, *Figure 3f*). Yet, only by compromising resolution for speed (and thus volume) of imaging, the mapping of large, potentially even whole-brain connectomes is becoming plausible (*Figure 3f*). Therefore, it was essential to obtain automated synapse detection for EM data that is of lower resolution and scalable to such volumes. The fact that SynEM also outperforms state-of-the-art methods on high-resolution anisotropic 3D EM data (*Figure 3f*, *Roncal et al., 2015*) indicates that our approach of segmentation-based interface classification has merits in a wider range of 3D EM data modalities.

In addition to high image resolution, recently proposed special fixation procedures that enhance the extracellular space in 3D EM data (*Pallotto et al., 2015*) are reported to simplify synapse detection for human annotators. In such data, direct touch between neurites has a very high predictive power for the existence of a (chemical or electrical) synapse, since otherwise neurite boundaries are separated by extracellular space. Thus, it is expected that such data also substantially simplifies automated synapse detection. The advantage of SynEM is that it achieves fully automated synapse detection in conventionally stained and fixated 3D EM data, in which neurite contact is most frequent at non-synaptic sites. Such data are widely used, and acquiring such data does not require special fixation protocols.

Finally, our approach to selectively classify interfaces of inhibitory axons (*Figure 5e*, *Figure 5—figure supplement 1*) requires discussion. So far, the classification of synapses into inhibitory (symmetric) vs. excitatory (asymmetric) was carried out for a given single synapse, often in single cross sections of single synapses (e.g. *Colonnier, 1968*). With the increasing availability of large-scale 3D EM datasets, however, synapse types can be defined based on multiple synapses of the same axon (e.g. *Kasthuri et al., 2015*). In the case of a dataset sized a cubic millimeter of cortical tissue, most axons of interneurons will be fully contained in the dataset since most inhibitory neurons are local. Consequently, the classification of single synapses can be replaced by the assignment of synapses to the respective axon; the type of axon is then inferred from the neurons' somatic and dendritic

features. Even for axons which are not completely contained in the dataset, the assignment to inhibitory or excitatory synaptic phenotypes can be based on dozens or hundreds rather than single synapses.

Together, SynEM resolves synapse detection for high-throughput cortical connectomics of mammalian brains, removing synapse detection as a bottleneck in connectomics. With this, SynEM renders the further acceleration of neurite reconstruction again the key challenge for future connectomic analysis.

## Materials and methods

### Annotation time estimates

Neuropil composition (*Figure 1b*) was considered as follows: Neuron density of 157,500 per mm$^3$ (*White and Peters, 1993*), axon path length density of 4 km per mm$^3$ and dendrite path length density of 1 km per mm$^3$ (*Braitenberg and Schüz, 1998*), spine density of about 1 per µm dendritic shaft length, with about 2 µm spine neck length per spine (thus twice the dendritic path length), synapse density of 1 synapse per µm$^3$ (*Merchán-Pérez et al., 2014*) and bouton density of 0.1–0.25 per µm axonal path length (*Braitenberg and Schüz, 1998*). Annotation times were estimated as 200–400 hr per mm path length for contouring, 3.7–7.2 h/mm path length for skeletonization (*Helmstaedter et al., 2011*, *2013*; *Berning et al., 2015*), 0.6 h/mm for flight-mode annotation (*Boergens et al., 2017*), 0.1 h/µm$^3$ for synapse annotation by volume search (estimated form the test set annotation) and an effective interaction time of 60 s per identified bouton for axon-based synapse search. All annotation times refer to single-annotator work hours, redundancy may be increased to reduce error rates in neurite and synapse annotation in these estimates (see *Helmstaedter et al., 2011*).

### EM image dataset and segmentation

SynEM was developed and tested on a SBEM dataset from layer 4 of mouse primary somatosensory cortex (dataset 2012-09-28_ex145_07x2, K.M.B. and M.H., unpublished data, see also *Berning et al., 2015*). Tissue was conventionally en-bloc stained (*Briggman et al., 2011*) with standard chemical fixation yielding compressed extracellular space (compare to *Pallotto et al., 2015*).

The image dataset was volume segmented using the SegEM algorithm (*Berning et al., 2015*). Briefly, SegEM was run using CNN 20130516T204040$_{8,3}$ and segmentation parameters as follows: $r_{se} = 0$; $\theta_{ms} = 50$; $\theta_{hm} = 0.39$; (see last column in*Table 2* in *Berning et al., 2015*). For training data generation, a different voxel threshold for watershed marker size $\theta_{ms} = 10$ was used. For test set and local connectome calculation the SegEM parameter set optimized for whole cell segmentations was used ($r_{se} = 0$; $\theta_{ms} = 50$; $\theta_{hm} = 0.25$, see *Table 2* in *Berning et al., 2015*).

### Neurite interface extraction and subvolume definition

Interfaces between a given pair of segments in the SegEM volume segmentation were extracted by collecting all voxels from the one-voxel boundary of the segmentation for which that pair of segments was present in the boundary's 26-neighborhood. Then, all interface voxels for a given pair of segments were linked by connected components, and if multiple connected components were created, these were treated as separate interfaces. Interface components with a size of 150 voxels or less were discarded.

To define the subvolumes around an interface used for feature aggregation (*Figure 2b*), we collected all voxels that were at a maximal distance of 40, 80 and 160 nm from any interface voxel and that were within either of the two adjacent segments of the interface. The interface itself was also considered as a subvolume yielding a total of 7 subvolumes for each interface.

### Feature calculation

Eleven 3-dimensional image filters with one to 15 instances each (*Table 1*) were calculated as follows and aggregated over the 7 subvolumes of an interface using 9 summary statistics, yielding 3224 features per directed interface. Image filters were applied to cuboids of size 548 × 548 × 268 voxels, each, which overlapped by 72,72 and 24 voxels in x,y and z dimension, respectively, to ensure that all interface subvolumes were fully contained in the filter output.

Gaussian filters were defined by evaluating the unnormalized 3d Gaussian density function

$$\hat{g}_\sigma(x,\ y,\ z) = \exp\left(-\frac{x^2}{2\sigma_x^2} - \frac{y^2}{2\sigma_y^2} - \frac{z^2}{2\sigma_z^2}\right)$$

at integer coordinates $(x,\ y,\ z) \in U = \{-f_x, -f_x-1,\ \dots\ f_x\} \times \{-f_y, -f_y-1,\ \dots\ f_y\} \times \{-f_z, -f_z-1,\ \dots\ f_z\}$ for a given standard deviation $\sigma = (\sigma_x, \sigma_y, \sigma_z)$ and a filter size $f = (f_x, f_y, f_z)$ and normalizing the resulting filter by the sum over all its elements

$$g_\sigma(x,\ y,\ z) = \frac{\hat{g}_\sigma(x,\ y,\ z)}{\sum_{(x',\ y',\ z') \in U} \hat{g}_\sigma(x',\ y',\ z')}.$$

First and second order derivatives of Gaussian filters were defined as

$$\frac{\partial}{\partial x} g_\sigma(x,\ y,\ z) = g_\sigma(x,\ y,\ z)\frac{-x}{\sigma_x^2},$$

$$\frac{\partial^2}{\partial x^2} g_\sigma(x,\ y,\ z) = g_\sigma(x,\ y,\ z)\left(\frac{x^2}{\sigma_x^2} - 1\right)\frac{1}{\sigma_x^2},$$

$$\frac{\partial}{\partial x}\frac{\partial}{\partial y} g_\sigma(x,\ y,\ z) = g_\sigma(x,\ y,\ z)\frac{xy}{\sigma_x^2\sigma_y^2}.$$

and analogously for the other partial derivatives. Normalization of $g_\sigma$ and evaluation of derivatives of Gaussian filters was done on U as described above. Filters were applied to the raw data I via convolution (denoted by $*$) and we defined the image's Gaussian derivatives as

$$I_x^\sigma(x,\ y,\ z) = I * \frac{\partial g_\sigma}{\partial x}(x,\ y,\ z),$$

$$I_{xy}^\sigma(x,\ y,\ z) = I * \frac{\partial^2 g_\sigma}{\partial x \partial y}(x,\ y,\ z)$$

and analogously for the other partial derivatives.

Gaussian smoothing was defined as $I * g_\sigma$.

Difference of Gaussians was defined as $(I * g_\sigma - I * g_{k\sigma})$, where the standard deviation of the second Gaussian filter is multiplied element-wise by the scalar k.

Gaussian gradient magnitude was defined as

$$\sqrt{I_x^\sigma(x,\ y,\ z)^2 + I_y^\sigma(x,\ y,\ z)^2 + I_z^\sigma(x,\ y,\ z)^2}.$$

Laplacian of Gaussian was defined as

$$I_{xx}^\sigma(x,\ y,\ z) + I_{yy}^\sigma(x,\ y,\ z) + I_{zz}^\sigma(x,\ y,\ z)$$

Structure tensor S was defined as a matrix of products of first order Gaussian derivatives, convolved with an additional Gaussian filter (window function) $g_{\sigma w}$:

$$S_{xy} = \left(I_x^{\sigma_D} I_y^{\sigma_D}\right) * g_{\sigma_w}$$

and analogously for the other dimensions, with standard deviation $\sigma_D$ of the image's Gauss derivatives. Since S is symmetric, only the diagonal and upper diagonal entries were determined, the eigenvalues were calculated and sorted by increasing absolute value.

The Hessian matrix was defined as the matrix of second order Gaussian derivatives:

$$H_{xy} = I_{xy}^\sigma,$$

and analogously for the other dimensions. Eigenvalues were calculated as described for the Structure tensor.

The local entropy feature was defined as

$$-\sum_{L\in\{0,\,...,\,255\}} p(L)log_2 p(L),$$

where p(L) is the relative frequency of the voxel intensity in the range {0, …, 255} in a given neighborhood U of the voxel of interest (calculated using the entropyfilt function in MATLAB).

Local standard deviation for a voxel at location (x, y, z) was defined by

$$\sqrt{\frac{1}{|U|-1}\sum_{(x',y',z')\in U} I(x',y',z') - \frac{1}{|U|(|U|-1)}\left(\sum_{(x',y',z')\in U} I(x',y',z')\right)^2},$$

for the neighborhood U of location (x, y, z) with |U| number of elements and calculated using MATLABs stdfilt function.

Sphere average was defined as the mean raw data intensity for a spherical neighborhood $U_r$ with radius r around the voxel of interest, with

$$U_r = \left\{(x,\,y,\,z)|x^2+y^2+(2z)^2\leq r^2\right\}\cap Z^3,$$

where $Z^3$ is the 3 dimensional integer grid; x,y,z are voxel indices; z anisotropy was approximately corrected.

The intensity/variance feature for voxel location (x, y, z) was defined as

$$\sum_{(x',y',z')\in U} I(x',y',z')^2 - \left(\sum_{(x',y',z')\in U} I(x',y',z')\right)^2,$$

for the neighborhood U of location (x, y, z).

The set of parameters for which filters were calculated is summarized in *Table 1*.

11 shape features were calculated for the border subvolume and the two 160 nm-restricted subvolumes, respectively. For this, the center locations (midpoints) of all voxels of a subvolume were considered. Shape features were defined as follows: The number of voxel feature was defined as the total number of voxels in the subvolumes. The voxel based diameter was defined as the diameter of a sphere with the same volume as the number of voxels of the subvolumes. Principal axes lengths were defined as the three eigenvalues of the covariance matrix of the respective voxel locations. Principal axes product was defined as the scalar product of the first principal components of the voxel locations in the two 160 nm-restricted subvolumes. Voxel based convex hull was defined as the number of voxels within the convex hull of the respective subvolume voxels (calculated using the convhull function in MATLAB).

## Generation of training and validation labels

Interfaces were annotated by three trained undergraduate students using a custom-written GUI (in MATLAB, *Figure 3—figure supplement 1*). A total of 40 non-overlapping rectangular volumes within the center 86 × 52 × 86 µm$^3$ of the dataset were selected (39 sized 5.6 × 5.6 × 5.6 µm$^3$ each and one of size 9.6 × 6.8 × 8.3 µm$^3$). Then, all interfaces within these volumes were extracted as described above. Interfaces with a center of mass less than 1.124 µm from the volume border were not considered. For each interface, a raw data volume of size (1.6 × 1.6 × 0.7–1.7) µm$^3$, centered on the center of mass of the interface voxel locations was presented to the annotator. When the center of mass was not part of the interface, the closest interface voxel was used. The raw data were rotated such that the second and third principal components of the interface voxel locations (restricted to a local surround of 15 x 15 × 7 voxels around the center of mass of the interface) defined the horizontal and vertical axes of the displayed images. First, the image plane located at the center of mass of the interface was shown. The two segmentation objects were transparently overlaid (*Figure 3—figure supplement 1*) in separate colors (the annotator could switch the labels

off for better visibility of raw data). The annotator had the option to play a video of the image stack or to manually browse through the images. The default video playback started at the first image. An additional video playback mode started at the center of mass of the interface, briefly transparently highlighted the segmentation objects of the interface, and then played the image stack in reverse order to the first plane and from there to the last plane. In most cases, this already yielded a decision. In addition, annotators had the option to switch between the three orthogonal reslices of the raw data at the interface location (*Figure 3—figure supplement 1*). The annotators were asked to label the presented interfaces as non-synaptic or synaptic. For the synaptic label, they were asked to indicate the direction of the synapse (see *Figure 3—figure supplement 1*). In addition to the annotation label interfaces could be marked as 'undecided'. Interfaces were annotated by one annotator each. The interfaces marked as undecided were validated by an expert neuroscientist. In addition, all synapse annotations were validated by an expert neuroscientist, and a subset of non-synaptic interfaces was cross-checked. Together, 75,383 interfaces (1858 synaptic, 73,525 non-synaptic) were labeled this way. Of these, the interfaces from eight label volumes (391 synaptic and 11906 non-synaptic interfaces) were used as validation set; the interfaces from the other 32 label volumes were used for training.

## SynEM classifier training and validation

The target labels for the undirected, augmented and directed label sets were defined as described in the Results (*Figure 3c*). We used boosted decision stumps (level-one decision trees) trained by the AdaBoostM1 (*Freund and Schapire, 1997*) or LogitBoost (*Friedman et al., 2000*) implementation from the MATLAB Statistical Toolbox (fitensemble). In both cases the learning rate was set to 0.1 and the total number of weak learners to 1500. Misclassification cost for the synaptic class was set to 100. Precision and recall values of classification results were reported with respect to the synaptic class. For validation, the undirected label set was used, irrespective of the label set used in training. If the classifier was trained using the directed label set then the thresholded prediction for both orientations were combined by logical OR.

## Test set generation and evaluation

To obtain an independent test set disjunct from the data used for training and validation, we randomly selected a volume of size $512 \times 512 \times 256$ voxels ($5.75 \times 5.75 \times 7.17$ $\mu m^3$) from the dataset that contained no soma or dominatingly large dendrite. One volume was not used because of unusually severe local image alignment issues which are meanwhile solved for the entire dataset. The test volume had the bounding box [3713, 2817, 129, 4224, 3328, 384] in the dataset. First, the volume was searched for synapses (see *Figure 1d*) in webKnossos (*Boergens et al., 2017*) by an expert neuroscientist. Then, all axons in the volume were skeleton-traced using webKnossos. Along the axons, synapses were searched (strategy in *Figure 1e*) by inspecting vesicle clouds for further potential synapses. Afterwards the expert searched for vesicle clouds not associated with any previously traced axon and applied the same procedure as above. In total, that expert found 335 potential synapses. A second expert neuroscientist used the tracings and synapse annotations from the first expert to search for further synapse locations. The second expert added eight potential synapse locations. All 343 resulting potential synapses were collected and independently assessed by both experts as synaptic or not. The experts labeled 282 potential locations as synaptic, each. Of these, 261 were in agreement. The 42 disagreement locations (21 from each annotator) were re-examined jointly by both experts and validated by a third expert on a subset of all synapses. 18 of the 42 locations were confirmed as synaptic, of which one was just outside the bounding box. Thus, in total, 278 synapses were identified. The precision and recall of the two experts in their independent assessment with respect to this final set of synapses was 93.6%, 94.6% (expert 1) and 97.9%, 98.9% (expert 2), respectively.

Afterwards all shaft synapses were labeled by the first expert and proofread by the second. Subsequently, the synaptic interfaces were voxel-labeled to be compatible with the methods by *Becker et al. (2012)* and *Dorkenwald et al. (2017)*. This initial test set comprised 278 synapses, of which 36 were labeled as shaft/inhibitory.

Next, all interfaces between pairs of segmentation objects in the test volume were extracted as described above. Then, the synapse labels were assigned to those interfaces whose border voxels

had any overlap with one of the 278 voxel-labeled synaptic interfaces. Afterwards, these interface labels were again proof-read by an expert neuroscientist. Finally, interfaces closer than 160 nm from the boundary of the test volume were excluded to ensure that interfaces were fully contained in the test volume. The final test set comprised 235 synapses out of which 31 were labeled as shaft/inhibitory. With this we obtained a high-quality test set providing both voxel-labeled synapses and synapse labels for interfaces, to allow the comparison of different detection methods.

For the calculation of precision and recall, a synapse was considered detected if at least one interface that had overlapped with the synapse was detected by the classifier (TPs); a synapse was considered missed if no overlapping interface of a given synapse was detected (FNs); and a detection was considered false positive (FP) if the corresponding interface did not overlap with any labeled synapse.

## Inhibitory synapse detection

The labels for inhibitory-focused synapse detection were generated using skeleton tracings of inhibitory axons. Two expert neuroscientists used these skeleton tracings to independently detect all synapse locations along the axons. Agreeing locations were considered synapses and disagreeing locations were resolved jointly by both annotators. The resulting test set contains 171 synapses. Afterwards, all SegEM segments of the consensus postsynaptic neurite were collected locally at the synapse location. For synapse classification all interfaces in the dataset were considered that contained one SegEM segment located in one of these inhibitory axons. Out of these interfaces all interfaces were labeled synaptic that were between the axon and a segment identified as postsynaptic. The calculation of precision and recall curves was done as for the dense test set (see above) by considering a synapse detected if at least one interface overlapping with it was detected by the classifier (TPs); a synapse was considered missed if no interface of a synapse was detected (FNs); and a detection was considered false positive (FP) if the corresponding interface did not overlap with any labeled synapse.

## Comparison to previous work

The approach of *Becker et al. (2012)* was evaluated using the implementation provided in Ilastik (*Sommer et al., 2011*). This approach requires voxel labels of synapses. We therefore first created training labels: an expert neuroscientist created sparse voxel labels at interfaces between pre- and postsynaptic processes and twice as many labels for non-synaptic voxels for five cubes of size 3.4 × 3.4 × 3.4 μm$^3$ that were centered in five of the volumes used for training SynEM. Synaptic labels were made for 115 synapses (note that the training set in *Becker et al. (2012)*) only contained 7–20 synapses). Non-synaptic labels were made for two training cubes first. The non-synaptic labels of the remaining cubes were made in an iterative fashion by first training the classifier on the already created synaptic and non-synaptic voxel labels and then adding annotations specifically for misclassified locations using Ilastik. Eventually, non-synaptic labels in the first two training cubes were extended using the same procedure.

For voxel classification all features proposed in (*Becker et al., 2012*) and 200 weak learners were used. The classification was done on a tiling of the test set into cubes of size 256 × 256 × 256 voxels (2.9 × 2.9 × 7.2 μm$^3$) with a border of 280 nm around each tile. After classification, the borders were discarded, and tiles were stitched together. The classifier output was thresholded and morphologically closed with a cubic structuring element of three voxels edge length. Then, connected components of the thresholded classifier output with a size of at least 50 voxels were identified. Synapse detection precision and recall rates were determined as follows: A ground truth synapse (from the final test set) was considered detected (TP) if it had at least a single voxel overlap with a predicted component. A ground truth synapse was counted as a false negative detection if it did not overlap with any predicted component (FN). To determine false positive classifications, we evaluated the center of the test volume (shrunk by 160 nm from each side to 484 × 484 × 246 voxels) and counted each predicted component that did not overlap with any of the ground truth synapses as false positive detection (FP). For this last step, we used all ground truth synapses from the initial test set, in favor of the *Becker et al. (2012)* classifier.

For comparison with (*Kreshuk et al., 2014*) the same voxel training data as for (*Becker et al., 2012*) was used. The features provided by Ilastik up to a standard deviation of 5 voxels for the voxel

classification step were used. For segmentation of the voxel probability output map the graph cut segmentation algorithm of Ilastik was used with label smoothing ([1, 1, 0.5] voxel standard deviation), a voxel probability threshold of 0.5 and graph cut constant of λ = 0.25. Objects were annotated in five additional cubes of size 3.4 × 3.4 × 3.4 µm$^3$ that were centered in five of the interface training set cubes different from the one used for voxel prediction resulting in 299 labels (101 synaptic, 198 non-synaptic). All object features provided by Ilastik were used for object classification. The evaluation on the test set was done as for (*Becker et al., 2012*).

For comparison with (*Dorkenwald et al., 2017*) six of the 32 training cubes used for interface classification with a total volume of 225 µm$^3$ were annotated with voxel labels for synaptic junctions, vesicle clouds and mitochondria. The annotation of vesicle clouds and mitochondria was done using voxel predictions of a convolutional neural network (CNN) trained on mitochondria, vesicle clouds and membranes. The membrane predictions were discarded and the vesicle clouds and mitochondria labels were first proofread by undergraduate students and then twice by an expert neuroscientist. The voxels labels for synaptic junctions were added by an expert neuroscientist based on the identified synapses in the interface training data. Overall 310 synapses were annotated in the training volume. A recursive multi-class CNN was trained on this data with the same architecture and hyperparameter settings as described in (*Dorkenwald et al., 2017*) using the ElektroNN framework. For the evaluation of synapse detection performance only the synaptic junction output was used. The evaluation on the test set was done as for (*Becker et al., 2012*) with a connected component threshold of 250 voxels.

## Evaluation on the dataset from *Kasthuri et al. (2015)*

The image data, neurite and synapse segmentation from (*Kasthuri et al., 2015*) hosted on openconnecto.me (kasthuri11cc, kat11segments, kat11synapses) was used (downloaded using the provided scripts at https://github.com/neurodata-arxiv/CAJAL). The segmentation in the bounding box [2432, 7552; 6656, 10112; 769, 1537] (resolution 1) was adapted to have a one-voxel boundary between segments by first morphologically eroding the original segmentation with a 3-voxel cubic structuring element and running the MATLAB watershed function on the distance-transform of the eroded segmentation on a tiling with cubes of size [1024, 1024, 512] voxels. Since the *Kasthuri et al. (2015)* segmentation in the selected bounding box was not dense, voxels with a segment id of zero in the original segmentation whose neighbors at a maximal distance of 2 voxels (maximum-distance) also all had segment ids zero were set to segment id zero in the adapted segmentation. All segments in the adapted segmentation that were overlapping with a segment in the original segmentation were set to the id of the segment in the original segmentation. The bounding box [2817, 6912; 7041, 10112; 897, 1408] of the resulting segmentation was tiled into non-overlapping cubes of [512, 512, 256] voxels. For all synapses in the synapse segmentation the pre- and postsynaptic segment of the synapse were marked using webKnossos (*Boergens et al., 2017*) and all interfaces between the corresponding segments at a maximal distance of 750 nm to the synapse centroid that were also overlapping with an object in the synapse segmentation were associated to the corresponding synapse and assigned a unique group id. Only synapses labeled as 'sure' in *Kasthuri et al. (2015)* were evaluated. All interfaces with a center of mass in the region ac3 with the bounding box [5472, 6496; 8712, 9736; 1000, 1256] were used for testing. All interfaces with a center of mass at a distance of at least 1 µm to ac3 were used for training if there was no interface between the same segment ids in the test set. Interfaces between the same segment ids as an interface in the test set were only considered for training if the distance to ac3 was above 2 µm. For feature calculation the standard deviation of Gaussian filters was adapted to the voxel size 6 × 6 × 30 nm of the data (i.e. s in *Table 1* was set to 12/6 in x- and y-dimension and 12/30 in z-dimension). The directed label set approach was used for classification. The calculation of precision recall rates was done as described above ('test set generation and evaluation').

## Pairwise connectivity model

The neuron-to-neuron connection recall was calculated assuming an empirical distribution p(n) of the number of synapses n between connected excitatory neurons given by published studies (see *Supplementary file 2*, *Feldmeyer et al., 1999*, *2002*, *2006*; *Frick et al., 2008*; *Markram et al., 1997*). For inhibitory connections we used a fixed value of 6 synapses (see *Supplementary file 3*,

*Koelbl et al., 2015*; *Hoffmann et al., 2015*; *Gupta et al., 2000*; *Markram et al., 2004*). We further assumed that the number of retrieved synapses is given by a binomial model with retrieval probability given by the synapse classifier recall $R_s$ on the test set:

$$P(k \geq \gamma_{nn} | R_s) = \sum_n Bin(k \geq \gamma_{nn} | n, R_s) p(n),$$

Where $\gamma_{nn}$ is the threshold on the number of synapses between a neuron pair to consider it as connected (see *Figure 5a*). This equates to the neuron-to-neuron recall: $R_{nn} = P(k \geq \gamma_{nn} | R_s)$.

To compute the neuron-to-neuron precision, we first calculated the expected number of false positive synapse detections (FP$_s$) made by a classifier with precision $P_s$ and recall $R_s$:

$$FP_s = \frac{(1 - P_s)}{P_s} R_s N_{syn}$$

where $N_{syn}$ is the total number of synapses in a dataset calculated from the average number of synapses per connected neuron pair $<n_{syn}>$ times the number of connected neuron pairs $N_{con}$ and $c_r$ is the connectivity ratio given by $N_{con}/N^2$ with N the number of neurons in the connectome.

We then assumed that these false positive synapse detections occur randomly and therefore are assigned to one out of $N^2$ possible neuron-to-neuron connections with a frequency $FP_s/N^2$.

We then used a Poisson distribution to estimate the number of cases in which at least $\gamma_{nn}$ FP$_s$ synapses would occur in a previously zero entry of the connectome, yielding a false positive neuron-to-neuron connection (FP$_{nn}$).

$$FP_{nn} = N^2(1 - c_r)Poi(x \geq \gamma_{nn} | FP_s/N^2).$$

Finally, the true positive detections of neuron-to-neuron connections in the connectome TP$_{nn}$ are given in terms of the neuron-to-neuron connection recall $R_{nn}$ by

$$TP_{nn} = N^2 c_r R_{nn}.$$

Together, the neuron-to-neuron connection precision $P_{nn}$ is given by

$$P_{nn} = \frac{TP_{nn}}{TP_{nn} + FP_{nn}} = \frac{c_r R_{nn}}{c_r R_{nn} + (1 - c_r)Poi(x \geq \gamma_{nn} | FP_s/N^2)}.$$

The connectivity ratio was set to $c_r$ = 0.2 (*Feldmeyer et al., 1999*) for excitatory and to 0.6 for inhibitory connections (*Gibson et al., 1999*; *Koelbl et al., 2015*).

## Local connectome

For determining the local connectome (*Figure 6*) between 104 pre- and 100 postsynaptic processes, we used 104 axonal skeleton tracings (traced at 1 to 5-fold redundancy) and 100 dendrite skeleton tracings. 10 axons were identified as inhibitory and are partially contained in the inhibitory test set. All volume objects which overlapped with any of the skeleton nodes were detected and concatenated to a given neurite volume. Then, all interfaces between pre- and postsynaptic processes were classified by SynEM. The area of each interface was calculated as in (*Berning et al., 2015*) and the total area of all contacts between all neurite pairs was calculated (*Figure 6b*). To obtain the weighted connectome C$_w$ (*Figure 6c*), we applied the SynEM scores threshold $\theta_{nn}$ (*Table 3*) for the respective presynaptic type (excitatory, inhibitory). Detected synaptic interfaces were clustered using hierarchical clustering (single linkage, distance cutoff 1500 nm) if the interfaces were between the same pre- and postsynaptic objects. To obtain the binary connectome C$_{bin}$ (*Figure 6d*) we thresholded the weighted connectome at the connectome threshold $\gamma_{nn}$ = 1 for excitatory and $\gamma_{nn}$ = 2 for inhibitory connections (*Table 3*). The overall number of synapses in the dataset was calculated by considering all interfaces above the score threshold for the best single synapse performance ($\theta_s$) as synaptic. To obtain the final synapse count the retrieved synaptic interfaces were clustered using hierarchical clustering with single linkage and a distance cutoff between the centroids of the interfaces of 320.12 nm (this distance cutoff was obtained by optimizing the synapse density prediction on the test set).

## Axon-spine interface area comparison

For the evaluation of axon-spine interface area (ASI) all spine synapses in the test set were considered for which SynEM had detected at least one overlapping neurite interface (using $\theta_s$ for spine synapses, *Figure 3e*). The ASI of a detected synapse was calculated by summing the area of all interfaces between segmentation objects that overlapped with the synapse. For comparison to ASI distributions obtained at higher imaging resolution in a recent study (spontaneous wake (SW) and enforced wake (EW) conditions reported in Table S1 in *de Vivo et al. (2017)*), it was assumed that the ASI distributions are lognormal (see Figure 2B in *de Vivo et al., 2017*). Two-sample two-tailed t-tests were performed for comparing the natural logarithmic values of the SynEM-detected ASI from the test set (log ASI $-1.60 \pm 0.74$, n = 181; mean $\pm$ s.d.) with the lognormal distributions for SW and EW from *de Vivo et al. (2017)*, (log ASI $-1.56 \pm 0.83$, n = 839, SW; $-1.59 \pm 0.81$, n = 836, EW; mean $\pm$ s.d.), p=0.5175 (SW) and p=0.8258 (EW).

## Code and data availability

All code used to train and run SynEM are available as source code and also at https://gitlab.mpcdf.mpg.de/connectomics/SynEM under the MIT license. A copy is available at https://github.com/elifesciences-publications/SynEM. To run SynEM, please follow instructions in the readme.md file. Data used to train and evaluate SynEM is available at http://synem.brain.mpg.de.

## Acknowledgements

We thank Jan Gleixner for first test experiments on synapse detection and fruitful discussions in an early phase of the project, Alessandro Motta for comments on the manuscript, Christian Guggenberger for excellent support with compute infrastructure, Raphael Jacobi, Raphael Kneissl, Athanasia Natalia Marahori, and Anna Satzger for data annotation and Elias Eulig, Robin Hesse, Martin Schmidt, Christian Schramm and Matej Zecevic for data curation. We thank Heiko Wissler and Dalila Rustemovic for support with illustrations.

## Additional information

### Funding

| Funder | Grant reference number | Author |
| --- | --- | --- |
| Max-Planck Society | Open access funding | Benedikt Staffler<br>Manuel Berning<br>Kevin M Boergens<br>Anjali Gour<br>Moritz Helmstaedter |

The funders had no role in study design, data collection and interpretation, or the decision to submit the work for publication.

### Author contributions

BS, Data curation, Software, Validation, Visualization, Methodology, Writing—original draft, Writing—review and editing; MB, Software, Methodology, Writing—review and editing; KMB, Validation, Writing—review and editing, Dataset acquisition; AG, Data curation, Validation, Writing—review and editing; PvdS, Supervision, Methodology, Writing—review and editing; MH, Conceptualization, Supervision, Validation, Methodology, Writing—original draft, Writing—review and editing

### Author ORCIDs

Benedikt Staffler, http://orcid.org/0000-0002-7383-305X
Manuel Berning, http://orcid.org/0000-0002-3679-8363
Moritz Helmstaedter, http://orcid.org/0000-0001-7973-0767

## Ethics

Animal experimentation: All animal experiments were performed in accordance with the guidelines for the Use of Laboratory Animals of the Max Planck Society and approved by the local authorities Regierungspräsidium Oberbayern, AZ 55.2-1-54-2532.3-103-12.

## Additional files

### Supplementary files

• Supplementary file 1. (Table ) 1 Overview of methods for automated synapse detection. Res. Fac: Image voxel volume of SBEM data used in this study relative to the voxel volume in the reported studies. Note that most studies employ data of substantially higher image resolution.

• Supplementary file 2. (Table) 2 Number of synapses between connected neurons obtained from published studies of paired recordings of excitatory neurons in rodent cortex. These distributions were used in *Figure 5* for prediction of connectome precision and recall.

• Supplementary file 3. (Table) 3 Number of synapses between connected neurons obtained from published studies of paired recordings of inhibitory neurons in rodent cortex.

• Supplementary file 4. Synapse gallery. Document describing the criteria by which synapses in 3D SBEM data were detected by human expert annotators. These criteria are exemplified for synapses from the test set of the SynEM classifier.

### Major datasets

The following dataset was generated:

| Author(s) | Year | Dataset title | Dataset URL | Database, license, and accessibility information |
| --- | --- | --- | --- | --- |
| Boergens K, Berning M, Staffler B, Gour A, Helmstaedter M | 2017 | SBEM data from mouse S1 cortex for SynEM development and validation | https://synem.rzg.mpg.de/webdav/ | Publicly available via Max Planck Computing and Data Facility |

The following previously published dataset was used:

| Author(s) | Year | Dataset title | Dataset URL | Database, license, and accessibility information |
| --- | --- | --- | --- | --- |
| Kasthuri N, Hayworth KJ, Berger DR, Schalek RL, Conchello JA, Knowles-Barley S, Lee D, Vázquez-Reina A, Kaynig V, Jones TR, Roberts M, Morgan JL, Tapia JC, Seung HS, Roncal WG, Vogelstein JT, Burns R, Sussman DL, Priebe CE, Pfister H, Lichtman JW | 2015 | Data from: Saturated Reconstruction of a Volume of Neocortex. | https://github.com/neurodata-arxiv/CAJAL | Publicly accessible via API (see 10.1016/j.cell.2015.06.054) |

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
