## [Decision Letter]

Thank you for submitting your article "SynEM: Automated synapse detection for connectomics" for consideration by *eLife*. Your article has been reviewed by two reviewers and the evaluation has been overseen by Jeremy Nathans as the Reviewing Editor and a Senior Editor..

As you will see, the reviewers were impressed with the importance and novelty of your work, but they also have a number of critiques/comments/suggestions that we think would improve the study and the manuscript.

I am including the unedited reviews at the end of this letter. The comments of reviewer #2 regarding generalizability (point 7) could represent a substantial amount of work. We suggest that you consider consolidating points 6 and 7 in reviewer #2's comments and compare the performance between SynEM, Dorkenwald et al., 2017, Roncal et al., 2014, and Becker et al., 2012 to at least one additional 3D EM dataset (as well as reporting Dorkenwald et al., 2017 and Roncal et al., 2014 performance on the first). The results would help to clarify SynEM's performance and applicability. You should consider reviewer #2's point 8 as optional.

In sum, we appreciate that the reviewers' comments cover a broad range of suggestions for improving the manuscript. Please use your best judgment in deciding which of these can be accommodated in a reasonable period of time.

Reviewer #1:

The manuscript by Staffler et al. describes a method for automatically detecting synapses in electron microscopy images collected using the block face scanning microscope. The authors report a high precision of 97%.

The appearance of this work is timely. An algorithm that is able to accurately detect synapses in electron micrographs with a lower z resolution than focussed ion beam images will be of interest to a number of groups. As the authors state, the method is scalable for larger sets of images, so should have a broad appeal.

This work is well presented and gives a thorough evaluation of the algorithms performance, as well as the comparison with other methods.

I have only a few relatively minor comments about the content of the paper. However, my major reservation concerns its suitability for this type of journal. The method is clearly tested on a very specific set of images, and its performance appears impressive, but beyond this there is little insight about the revealed connectivity. This is clearly a technically important piece of work, but I wonder if it should not be suited to a more specialist journal.

On a more specific note, I was confused about the testing of the performance on the different types of synapses. There is a description of how synapses along inhibitory axons were identified, and the success was high. However, does this mean that it is able to distinguish the contacts, but not classify between excitatory and inhibitory? How does the overall success for identifying synapses in the entire volume account for the two types of synapse? I presume from the description that it is simply identifying a contact, and no distinction is made. This needs to be clarified. It is perhaps a minor point, but nevertheless it is not explicitly given. It is also not clear as to whether the program is able to distinguish those contacts that are found on the dendritic shaft as opposed to those on the spine. Contextual information is being used from these different sites of contact, but are they given in the final result?

Reviewer #2:

Staffler and colleagues present SynEM, a tool for automated chemical synapse inference in volumetric electron microscopy data of cortical neuropil. The authors report SynEM: (1) provides 97% precision and recall of binary cortical connectivity without user interaction, (2) scales to large volumes, and (3) possibly to whole-brain datasets. SynEM is a natural extension of the group's analysis workflow for connectomics. Relying on volumetric neurite segmentation following manual skeletonization with SegEM (Berning et al., Neuron 2015), SynEM performs synapse inference by classifying interfaces between neurites based on spatial, texture, and shape features. The authors apply SynEM to their SBEM dataset from the mouse cortex and present the extracted connectivity as adjacencies for a local connectome in L4 of barrel cortex.

SynEM could be of potential useful to the field, however, there are several important concerns about its potential presented here:

1) Is SBEM + SegEM + SynEM comprehensively detecting structural connectivity? This is a fundamental advantage of EM. Estimating 1 synapse/μm^3^ (Merchan-Perez et al., 2014), in the authors' 86 x 86 x 52 μm^3^ volume (Figure 4), one would expect 384592 synapses. In the reported local cortical connectome (subsection “Local cortical connectome”), SynEM detected 813 synapses over 531 connections. The number of detected synapses are 3 orders of magnitude less than expected. What might explain this? Either A. Only a subset of the data was used; B. The neuropil is not representative of estimates from the literature; C. The SynEM workflow is tuned to detect a subset of synapses, perhaps biased toward "large" synapses, also influencing precision-recall rates; or D. The dataset has insufficient information to detect many cortical synapses.

If (A), please clarify in the text/methods and let readers know if the synapse density is as expected, or why not. Given that the segmentation with SegEM was volumetric, a comprehensive connectome within the volume should be possible. If not, why did the authors choose only 104 axons and 100 postsynaptic processes?

If (B), please explain why neuropil in this dataset is not representative.

If (C) or (D), this is, of course, would significantly limit the utility of these approaches for the field as the results would not accurately represent the structural connectivity of neuronal networks.

2) Ground truth: synapse numbers were generated by viewing post-segmentation post-border detection volumes. How many synapses were lost during these preprocessing steps? A comparison against ground-truth synapse data generated using an independent method is necessary to judge the overall accuracy of the method in detecting synapses within a volume (see also point 1 above).

3) Are precision and recall rates elevated by large synapses? The distribution of synapse sizes reported in the Supplemental Synapse Gallery (subsection “SynEM workflow and training data”) appears heavily weighted toward larger interfaces (median > 0.1 μm^2^) compared to previous reports. PSD areas have reported median values of < 0.05 μm^2^ between hippocampal pyramidal cells (Harris and Stevens, J Neurosci 1989; Bartol et al., *eLife* 2015) and appear to be even smaller for both putative thalamic and non-thalamic synapses in L4 of mouse barrel cortex (Bopp et al., J Neurosci 2017). Thus, synapses < 0.1 μm^2^ are unlikely to be a small fraction of the actual population as asserted (Supplemental Synapse Gallery). Rather, it is worrisome that this approach may be missing synapses detectable with other methods (points 1-2), leaving and enriching for synapses easily detected by SynEM.

4) This reader found it difficult to interpret how ground-truth neuron-to-neuron connections were generated. It appears that neuron-to-neuron connectivity precision was calculated using several values taken from the literature (including pairwise connectivity rate and mean number of synapses per connection, subsection “SynEM for connectomes”). An additional assumption in their pairwise connectivity model is that connectivity is random and independent. There is increasing evidence that this assumption is not supported in the rodent cortex (e.g. Song et al., Plos Biol 2005; Lefort et al., Neuron 2009; Ko et al., Nature 2011; Perin et al., PNAS 2011). Is the cost of ground-truthing the reason these numbers were not measured from the dataset used to evaluate SynEM? This is an important value to get correct given the authors' claims.

5) SegEM errors influencing SynEM: SynEM requires a segmented volume (generated here by SegEM). A fuller description of how segmentation errors affect SynEM performance (more than just a few example images) would allow the reader to determine what segmentation results are suitable for application of SynEM (this may also inform point 1 above). Description of how errors in assignment of extracellular space affect SynEM performance would allow the authors support their claim that space-preserving EM preparations would simplify synapse detection.

6) Comparisons and references to relevant and available state-of-the-art tools: To provide evidence that SynEM provides a substantial methodological advance with the new potential to facilitate difficult or intractable experiments, the authors should provide fair comparisons of SynEM to multiple the state-of-the-art approaches. The authors do well to identify a set ([Supplementary-material SD12-data]), but appear to only apply and report their application of Becker et al., 2012 (perhaps to represent it and the family of approaches from the Hamprecht group?) to their data in the main Figures. Several others are missing:

Dorkenwald, S. et al. Automated synaptic connectivity inference for volume electron microscopy. Nat Methods (2017).

Perez, A.J. et al. A workflow for the automatic segmentation of organelles in electron microscopy image stacks. Front. Neuroanat. (2014).

Roncal, W.G. et al. VESICLE: volumetric evaluation of synaptic interfaces using computer vision at large scale. Preprint available at https://arxiv. org/abs/1403.3724 (2014).

Dorkenwald, 2017 was published just recently so its omission is perhaps unsurprising; however, Dorkenwald, 2017 and Roncal, 2014 are particularly important with a similar aims of large-scale synapse inference and may provide competitive or better performance compared to SynEM than the other approaches the authors list.

The authors should also show the precision-recall curves for the other methods (perhaps the 3 best performers from [Supplementary-material SD12-data]; Dorkenwald, 2017; Perez, 2014; and Roncal, 2014) tuned to their dataset. This would be an appropriate supplement to Figure 2.

7) Generalizability: A major impediment to the EM connectomics field has been that analysis workflows have been highly specific to particular dataset types. It would be of substantially more value if SynEM's utility is demonstrated across different 3D EM dataset types. The abstract states: "we report SynEM, a method for automated detection of synapses from conventionally en-bloc stained 3D electron microscopy image stacks"

As the authors are aware, there are multiple approaches to generate 3D EM datasets. To provide evidence for the utility across 3D EM image stacks, the authors should apply SynEM to other EM datasets. Ideally, datasets from 3D EM methods (Briggman and Bock, 2012) other than SBEM. This approach (1) would demonstrate generalizability; (2) could provide a better understanding of dataset parameters influencing SynEM and automated segmentation performance more broadly (it is the lower resolution of their dataset that benefits most from SynEM, on the other hand the authors may discover that SynEM is an overall outperformer?); and (3) benefits from the authors' knowledge on how to best tune the SynEM pipeline as compared to others' workflows (a potential problem with comparing others' approaches on one's own data). This should be straightforward. Several public datasets exist of ATUM-SEM (e.g. https://neurodata.io/data/kasthuri15), ssTEM (eg. https://neurodata.io/data/bock11), FIB-SEM (e.g. http://cvlab.epfl.ch/data/em), and other SBEM data. Alternatively, the authors could modify their description of scope, narrowing the utility of their approach to SBEM. In this case, one would, still want to see the workflow evaluated on at least one other independent dataset.

8) Directly test the claim of lower resolution performance: The authors propose that better segmentation tools are needed for higher-imaging throughput, lower-resolution datasets and that SynEM's performance is particularly useful for such data. In addition to testing performance on other dataset types (point 7), it should be straightforward for the authors to compare performance on their 6 nm data (Figure 1). The authors should demonstrate the difference in performance on 6 compared to 12 nm data. If the authors are interested in imaging fast at lowest resolution possible to accurately extract connectomes, they should also decimate their 12 nm data to lower resolutions and examine where SynEM performance falls off in as a function of resolution.

9) Scalability: The authors strongly pitch that to analyze a 1 mm^3^ of cortex, approaches must be developed to accurately detect billions of synapses. In the abstract, the authors also assert that SynEM may plausibly scale to whole-brain datasets. It is unclear synEM the tool up to this task. As alluded to in (point 1) above, the synapse detection rates are surprisingly low. Moreover, there is little detail about how the approach scales to much larger volumes in terms of compute time and resources or other aspects of effort in the pipeline. Thus, it is difficult for this reviewer to estimate the cost and effort of implementing this workflow at such scales.

10) Terminology: binary vs. binary and undirected. To this reviewer, the use of 'binary' should be clarified and made consistent throughout the text. At the synapse level, the authors use binary to classify appositions as synapses or not, irrespective of direction.

Subsection “SynEM workflow and training data”. The SynEM score was then thresholded to obtain an automated binary classification of interfaces into synaptic / non-synaptic (θ in Figure 2).

Subsection “SynEM workflow and training data” and Figure 2 legend Initially, we interpreted the annotator's labels in a binary fashion: irrespective of synapse direction, the label was interpreted as synaptic (and non-synaptic otherwise, Figure 2, "Binary")

Whereas, at the connection level (all synapses making up the connectivity between a neuron pair) binary is used for whether or not there is at least γ synapses. These are, however, are directed connections.

Subsection “SynEM for connectomes”.We assume that the goal is a binary connectome containing the information whether pairs of neurons are connected or not.

Subsection “SynEM for connectomes”. “binary connectomes by considering all neuron pairs with at least γnn synapses as connected”

Convention in the field is for 'binary' to be used for unweighted connectivity, not to describe an undirected synapse. In descriptions of connectivity this reviewer would suggest using 'undirected-binary', 'directed-binary', 'undirected-weighted', or 'directed-weighted' not just 'binary' across the text.

---

## [Author Response]

*Reviewer #1:*

*I have only a few relatively minor comments about the content of the paper. However, my major reservation concerns its suitability for this type of journal. The method is clearly tested on a very specific set of images, and its performance appears impressive, but beyond this there is little insight about the revealed connectivity. This is clearly a technically important piece of work, but I wonder if it should not be suited to a more specialist journal.*

We addressed the potential concern of a “specific set of images”, also following the suggestions by reviewer 2, and tested SynEM on a different 3D EM dataset obtained using ATUM-SEM (Kasthuri et al., 2015). SynEM also outperforms the state-of-the art method on this dataset (Roncal et al., 2015, see updated Figure 3 and new Figure 3—figure supplement 4).

As to the suitability of a methods paper for *eLife*: we submitted this manuscript in the tools and resources section of *eLife*, hoping that this would be the proper place for a methodological manuscript. Naturally, the methodological manuscript is not primarily about novel biological insights, but about the demonstration that the method will be suited to obtain such in the future. We hope this clarifies this concern.

*On a more specific note, I was confused about the testing of the performance on the different types of synapses. There is a description of how synapses along inhibitory axons were identified, and the success was high. However, does this mean that it is able to distinguish the contacts, but not classify between excitatory and inhibitory? How does the overall success for identifying synapses in the entire volume account for the two types of synapse? I presume from the description that it is simply identifying a contact, and no distinction is made. This needs to be clarified. It is perhaps a minor point, but nevertheless it is not explicitly given. It is also not clear as to whether the program is able to distinguish those contacts that are found on the dendritic shaft as opposed to those on the spine. Contextual information is being used from these different sites of contact, but are they given in the final result?*

The process of synapse detection and classification is as follows: SynEM first determines all contacts (direct touches) between neurites. It then classifies the contact as synaptic or not, irrespective of synapse type (results reported in Figure 3). This data thus includes excitatory and inhibitory synapses at their relative prevalence in the neuropil (typically about 80-85% excitatory vs 15-20% inhibitory). In addition, we report SynEM performance for synapses onto spines, only (Figure 3, subsection “SynEM evaluation”), which is slightly better than for all synapses together. We then investigated whether given an axon that is known to be inhibitory, what is the SynEM detection performance for all outgoing neurite contacts of this axon. This data is reported in Figure 5—figure supplement 1 and subsection “SynEM for connectomes”, and yields an inhibitory synapse detection well suited for fully automated connectome mapping.

Why is this a reasonable approach? In large-scale 3D EM data, as we are analyzing here, the nature of the axon is mostly directly identified by the exit from a local interneuron cell body. For axons without a cell body in the dataset, the nature of the axon is identified by the statistics of its synapses onto spines or shafts. Therefore, the distinction between excitatory and inhibitory synapses does not anymore have to rely on single synaptic interfaces, as was historically the case, but can exploit all information from an entire local axon.

We tried to make this clearer in the manuscript (Discussion section).

It should also be noted that the comparison to published methods for synapse detection is reported for all synapses (i.e. excitatory and inhibitory synapses lumped together at their relative prevalence in the data), see Figure 3, since this is the synapse detection performance as it is reported by these studies (Mishchenko et al., 2010, Kreshuk et al., 2011, Becker et al., 2012, Kreshuk et al., 2014, Roncal et al., 2015, Dorkenwald et al., 2017).

*Reviewer #2:*

*[…] 1) Is SBEM + SegEM + SynEM comprehensively detecting structural connectivity? This is a fundamental advantage of EM. Estimating 1 synapse/μm^3^ (Merchan-Perez et al., 2014), in the authors' 86 x 86 x 52 μm^3^ volume (Figure 4), one would expect 384592 synapses. In the reported local cortical connectome (subsection “Local cortical connectome”), SynEM detected 813 synapses over 531 connections. The number of detected synapses are 3 orders of magnitude less than expected. What might explain this? Either A. Only a subset of the data was used; B. The neuropil is not representative of estimates from the literature; C. The SynEM workflow is tuned to detect a subset of synapses, perhaps biased toward "large" synapses, also influencing precision-recall rates; or D. The dataset has insufficient information to detect many cortical synapses.*

*If (A), please clarify in the text/methods and let readers know if the synapse density is as expected, or why not. Given that the segmentation with SegEM was volumetric, a comprehensive connectome within the volume should be possible. If not, why did the authors choose only 104 axons and 100 postsynaptic processes?*

*If (B), please explain why neuropil in this dataset is not representative.*

*If (C) or (D), this is, of course, would significantly limit the utility of these approaches for the field as the results would not accurately represent the structural connectivity of neuronal networks.*

The reviewer addresses an important point: the exhaustive detection of synapses. Our local connectome reported as an example in Figure 4 (now Figure 6) was a sparse local connectome, which we should have more clearly pointed out (between 104 axons and 100 dendrites, subsection “SynEM for connectomes”). The overall recall of SynEM is reported in Figure 3 and is around 90%.

To address the specific concern of the reviewer, we have run SynEM on half of the entire cortex dataset (volume: 192296 µm^3^), yielding 195.644 synapse detections, in fact very close to the reviewers’ estimate. We have added this result to the main text to help clarify this issue also for other readers (subsection “Frequency and size of automatically detected synapses”).

*2) Ground truth: synapse numbers were generated by viewing post-segmentation post-border detection volumes. How many synapses were lost during these preprocessing steps? A comparison against ground-truth synapse data generated using an independent method is necessary to judge the overall accuracy of the method in detecting synapses within a volume (see also point 1 above).*

For training data generation, the described more efficient tool was used (subsection “SynEM workflow and training data”). For test set generation, however, the test volume was meticulously volume searched for synapses using only our 3D data viewer webKnossos (Boergens et al., 2017; as described in Material and methods). Therefore our test results (which all interpretations are based on) constitute such an independent method of expert synapse detection (by 3 experts, as described in Material and methods).

*3) Are precision and recall rates elevated by large synapses? The distribution of synapse sizes reported in the Supplemental Synapse Gallery (subsection “SynEM workflow and training data”) appears heavily weighted toward larger interfaces (median > 0.1 μm^2^) compared to previous reports. PSD areas have reported median values of < 0.05 μm^2^ between hippocampal pyramidal cells (Harris and Stevens, J Neurosci 1989; Bartol et al., eLife 2015) and appear to be even smaller for both putative thalamic and non-thalamic synapses in L4 of mouse barrel cortex (Bopp et al., J Neurosci 2017). Thus, synapses < 0.1 μm^2^ are unlikely to be a small fraction of the actual population as asserted (Supplemental Synapse Gallery). Rather, it is worrisome that this approach may be missing synapses detectable with other methods (points 1-2), leaving and enriching for synapses easily detected by SynEM.*

We acknowledge the reviewer’s concern about the dependence between synapse size and automated detection. In the synapse gallery, we reported the *contact size*, i.e. the axo-dendritic (spine) area, *not* the PSD area, that the reviewer is referring to. Figure 8 shows the distribution of PSD area in our data and in Kasthuri et al., 2015. Our data is even smaller than Kasthuri et al., and in the range of Harris et al. (which in turn is smaller than Kasthuri et al).

Author response image 1.Overview over published PSD area distribution (Harris and Stevens, 1989; Bartol et al., 2015; Kasthuri et al., 2015; Bopp et al., 2017) in comparison to the SynEM test set PSD area distribution.Ranges as specified in the respective paper (Harris and Stevens, 1989) or estimated from the figures (Bartol et al., 2015; Bopp et al., 2017). The PSD area distribution for (Kasthuri et al., 2015) was calculated using the same method as for the SynEM test set from the synapse segmentation published in (Kasthuri et al., 2015).**DOI:**
http://dx.doi.org/10.7554/eLife.26414.035

As to axon-spine interfaces, we report these in new Figure 7 in comparison to a recent connectomics study that primarily analyzed axon-spine interface area (de Vivo et al., 2017).

Together it is evident that our detection is not biased towards larger synapses than previously reported.

*4) This reader found it difficult to interpret how ground-truth neuron-to-neuron connections were generated. It appears that neuron-to-neuron connectivity precision was calculated using several values taken from the literature (including pairwise connectivity rate and mean number of synapses per connection, subsection “SynEM for connectomes”). An additional assumption in their pairwise connectivity model is that connectivity is random and independent. There is increasing evidence that this assumption is not supported in the rodent cortex (e.g. Song et al., Plos Biol 2005; Lefort et al., Neuron 2009; Ko et al., Nature 2011; Perin et al., PNAS 2011). Is the cost of ground-truthing the reason these numbers were not measured from the dataset used to evaluate SynEM? This is an important value to get correct given the authors' claims.*

The reviewer discusses our estimates of SynEM performance for neuron-to- neuron connections (old Figure 3, new Figure 5). The reviewer is correct that we use the published data on the number of synapses per connection for our estimates. Beyond that, however we do not make any assumption on the particular structure of the neuronal connectivity (such as in the studies cited by the reviewer). We only estimate the effect of the fact that a given neuron-to-neuron connection is established via multiple, not one synapse. Our estimate does therefore not include an assumption about higher-order connectivity. It does assume that the detection of synapses between a pair of neurons is independent, since these synapses are often on different branches of the same pre-and postsynaptic neuron (see e.g. Feldmeyer et al., 1999). To briefly address the effect of this assumption, we ran our estimates for an inhomogeneous distribution of synapse sizes over given neuron-to-neuron connections. Assume for example that 20% of connections are made by only the 20% smallest synapses, and vice versa. Then we analyzed the performance of SynEM on these smaller synapses vs. larger synapses in the test set, and re-ran our model for connectome error prediction. The effect is reported in Figure 9, changes of less than 0.5% in predicted precision and recall. This illustrates that while our assumption of independence between the 2-10 synapses of a given neuron-to-neuron connection is a simplification, the effect on the performance prediction is modest.

Author response image 2.Estimated neuron-to-neuron recall and precision if synapses are assumed to be retrieved independently by the classifier (red) and assuming that 20% of the connections are made exclusively by the 20% smallest synapses and vice versa (blue).**DOI:**
http://dx.doi.org/10.7554/eLife.26414.036

*5) SegEM errors influencing SynEM: SynEM requires a segmented volume (generated here by SegEM). A fuller description of how segmentation errors affect SynEM performance (more than just a few example images) would allow the reader to determine what segmentation results are suitable for application of SynEM (this may also inform point 1 above). Description of how errors in assignment of extracellular space affect SynEM performance would allow the authors support their claim that space-preserving EM preparations would simplify synapse detection.*

The reviewer raises a relevant point. We use a published SegEM-segmentation (Berning et al., 2015), whose properties are described and quantified there at great detail, including downloadable datasets, allowing a comparison to other segmentation approaches.

To address the particular question of whether the quality of SegEM affects SynEM performance, we locally corrected all SynEM classification errors (FPs and FNs) of the test set and reran the SynEM classification. In fact, 13 of 28 FNs were caused by over-split SegEM segments, similarly 9 of 27 FPs. This means that the remaining error rate of SynEM could be even smaller when a “perfect” volume segmentation can be used. We added this point to the manuscript (subsection “Remaining SynEM errors, feature importance, and computational feasibility”).

*6) Comparisons and references to relevant and available state-of-the-art tools: To provide evidence that SynEM provides a substantial methodological advance with the new potential to facilitate difficult or intractable experiments, the authors should provide fair comparisons of SynEM to multiple the state-of-the-art approaches. The authors do well to identify a set ([Supplementary-material SD12-data]), but appear to only apply and report their application of Becker et al., 2012 (perhaps to represent it and the family of approaches from the Hamprecht group?) to their data in the main Figures. Several others are missing:*

*Dorkenwald, S. et al. Automated synaptic connectivity inference for volume electron microscopy. Nat Methods (2017).*

*Perez, A.J. et al. A workflow for the automatic segmentation of organelles in electron microscopy image stacks. Front. Neuroanat. (2014).*

*Roncal, W.G. et al. VESICLE: volumetric evaluation of synaptic interfaces using computer vision at large scale. Preprint available at https://arxiv. org/abs/1403.3724 (2014).*

*Dorkenwald, 2017 was published just recently so its omission is perhaps unsurprising; however, Dorkenwald, 2017 and Roncal, 2014 are particularly important with a similar aims of large-scale synapse inference and may provide competitive or better performance compared to SynEM than the other approaches the authors list.*

*The authors should also show the precision-recall curves for the other methods (perhaps the 3 best performers from [Supplementary-material SD12-data]; Dorkenwald, 2017; Perez, 2014; and Roncal, 2014) tuned to their dataset. This would be an appropriate supplement to Figure 2.*

We compared to the methods available at the time of preparation of the manuscript (which excluded Dorkenwald et al., 2017), and only implemented full- scale comparisons for those methods that were best performing at the time (i.e. if a published method had been shown to be inferior to another published method, we chose the better one). This is documented in [Supplementary-material SD12-data]. Becker et al., 2012 was the best-performing competitor.

However, to address this issue, we have now added a comparison to Dorkenwald, 2017, which is itself better than any other published methods (their Figure 2D in Dorkenwald et al., 2017). Please find the results in new Figure 3, Figure 3—figure supplement 3. SynEM outperforms Dorkenwald et al., 2017 on our data. Furthermore, SynEM outperforms Roncal et al., 2015 on their data (see below reply to point 7). With this, all-way comparisons are available to the currently top performing tools.

*7) Generalizability: A major impediment to the EM connectomics field has been that analysis workflows have been highly specific to particular dataset types. It would be of substantially more value if SynEM's utility is demonstrated across different 3D EM dataset types. The abstract states: "we report SynEM, a method for automated detection of synapses from conventionally en-bloc stained 3D electron microscopy image stacks"*

*As the authors are aware, there are multiple approaches to generate 3D EM datasets. To provide evidence for the utility across 3D EM image stacks, the authors should apply SynEM to other EM datasets. Ideally, datasets from 3D EM methods (Briggman and Bock, 2012) other than SBEM. This approach (1) would demonstrate generalizability; (2) could provide a better understanding of dataset parameters influencing SynEM and automated segmentation performance more broadly (it is the lower resolution of their dataset that benefits most from SynEM, on the other hand the authors may discover that SynEM is an overall outperformer?); and (3) benefits from the authors' knowledge on how to best tune the SynEM pipeline as compared to others' workflows (a potential problem with comparing others' approaches on one's own data). This should be straightforward. Several public datasets exist of ATUM-SEM (e.g. https://neurodata.io/data/kasthuri15), ssTEM (eg. https://neurodata.io/data/bock11), FIB-SEM (e.g. http://cvlab.epfl.ch/data/em), and other SBEM data. Alternatively, the authors could modify their description of scope, narrowing the utility of their approach to SBEM. In this case, one would, still want to see the workflow evaluated on at least one other independent dataset.*

We disagree with the conceptual notion of an all-interchangeable methodological pipeline (in the end, the goal is to go from brain tissue to connectivity data – if tailored tool chains are better than one-for-all, so be it) – but this is a strategic discussion beyond the scope of the manuscript.

To address the concern about generalizability to other 3D EM data, we followed the reviewer’s suggestion and applied SynEM to the ATUM-SEM data by Kasthuri et al., 2015. Please find the results in Figure 3—figure supplement 4, compared to Roncal et al., 2015, which was optimized for this data. Again, SynEM outperforms the current method.

With this, we hope to have addressed the concerns in points 6 and 7 about comparison to the latest best performing tool (Dorkenwald et al., 2017) and for other 3D EM data.

*8) Directly test the claim of lower resolution performance: The authors propose that better segmentation tools are needed for higher-imaging throughput, lower-resolution datasets and that SynEM's performance is particularly useful for such data. In addition to testing performance on other dataset types (point 7), it should be straightforward for the authors to compare performance on their 6 nm data (Figure 1). The authors should demonstrate the difference in performance on 6 compared to 12 nm data. If the authors are interested in imaging fast at lowest resolution possible to accurately extract connectomes, they should also decimate their 12 nm data to lower resolutions and examine where SynEM performance falls off in as a function of resolution.*

We agree that this would be an interesting extension, and in fact we do think that a bit lower resolution is still sufficient for automated synapse detection – but we would like to follow the editors’ judgement that this is not a required point for the revision.

*9) Scalability: The authors strongly pitch that to analyze a 1 mm^3^ of cortex, approaches must be developed to accurately detect billions of synapses. In the abstract, the authors also assert that SynEM may plausibly scale to whole-brain datasets. It is unclear synEM the tool up to this task. As alluded to in (point 1) above, the synapse detection rates are surprisingly low. Moreover, there is little detail about how the approach scales to much larger volumes in terms of compute time and resources or other aspects of effort in the pipeline. Thus, it is difficult for this reviewer to estimate the cost and effort of implementing this workflow at such scales.*

Of course SynEM has a very high recall as reported (Figure 3) – and detects the expected rate of synapses (see reply to point 1 above).

As to the computational load: SynEM has not been optimized yet for speed. But even with the current matlab code, synapse detection on a 1 mm^3^ dataset would run about 280 days on a mid-size computational cluster (extrapolated from 2.5 days run time for the cortex dataset reported here). This is well comparable to segmentation run time, and much less than the still required human interaction time. We added these estimates to the manuscript (subsection “Remaining SynEM errors, feature importance, and computational feasibility”).

*10) Terminology: binary vs. binary and undirected. To this reviewer, the use of 'binary' should be clarified and made consistent throughout the text. At the synapse level, the authors use binary to classify appositions as synapses or not, irrespective of direction.*

*Subsection “SynEM workflow and training data”. The SynEM score was then thresholded to obtain an automated binary classification of interfaces into synaptic / non-synaptic (θ in Figure 2).*

*Subsection “SynEM workflow and training data” and Figure 2 legend Initially, we interpreted the annotator's labels in a binary fashion: irrespective of synapse direction, the label was interpreted as synaptic (and non-synaptic otherwise, Figure 2, "Binary")*

*Whereas, at the connection level (all synapses making up the connectivity between a neuron pair) binary is used for whether or not there is at least γ synapses. These are, however, are directed connections.*

*Subsection “SynEM for connectomes”. We assume that the goal is a binary connectome containing the information whether pairs of neurons are connected or not.*

*Subsection “SynEM for connectomes”.“binary connectomes by considering all neuron pairs with at least γnn synapses as connected”*

*Convention in the field is for 'binary' to be used for unweighted connectivity, not to describe an undirected synapse. In descriptions of connectivity this reviewer would suggest using 'undirected-binary', 'directed-binary', 'undirected-weighted', or 'directed-weighted' not just 'binary' across the text.*

We appreciate this point by the reviewer and have amended the terminology accordingly (subsection “SynEM evaluation”, subsection “SynEM classifier training and validation” and Figure 3).